# A rapid simple point-of-care assay for the detection of SARS-CoV-2 neutralizing antibodies

Patthara Kongsuphol[1,8], Huan Jia[1,8], Hoi Lok Cheng [1,8], Yue Gu[2,8], Bhuvaneshwari D/O Shunmuganathan[2,8], Ming Wei Chen [3], Sing Mei Lim [1], Say Yong Ng[1], Paul Ananth Tambyah[4,5], Haziq Nasir[4], Xiaohong Gao [3], Dousabel Tay[6], Seunghyeon Kim[6], Rashi Gupta[2], Xinlei Qian[7], Mary M. Kozma[7], Kiren Purushotorman [2], Megan E. McBee [1], Paul A. MacAry[2,7,9 ✉], Hadley D. Sikes [1,6,9 ✉] & Peter R. Preiser [1,3,9 ✉]

### Abstract

**Background** Neutralizing antibodies (NAbs) prevent pathogens from infecting host cells. Detection of SARS-CoV-2 NAbs is critical to evaluate herd immunity and monitor vaccine efficacy against SARS-CoV-2, the virus that causes COVID-19. All currently available NAb tests are lab-based and time-intensive.

**Method** We develop a 10 min cellulose pull-down test to detect NAbs against SARS-CoV-2 from human plasma. The test evaluates the ability of antibodies to disrupt ACE2 receptor—RBD complex formation. The simple, portable, and rapid testing process relies on two key technologies: (i) the vertical-flow paper-based assay format and (ii) the rapid interaction of cellulose binding domain to cellulose paper.

**Results** Here we show the construction of a cellulose-based vertical-flow test. The developed test gives above 80% sensitivity and specificity and up to 93% accuracy as compared to two current lab-based methods using COVID-19 convalescent plasma.

**Conclusions** A rapid 10 min cellulose based test has been developed for detection of NAb against SARS-CoV-2. The test demonstrates comparable performance to the lab-based tests and can be used at Point-of-Care. Importantly, the approach used for this test can be easily extended to test RBD variants or to evaluate NAbs against other pathogens.

### Plain language summary

In response to infections, the human body produces proteins called antibodies. Neutralizing antibodies (NAbs) are one type of such proteins that are capable of inactivating the target, such as the SARS-CoV-2 virus that causes COVID-19. Monitoring levels of NAb allows us to understand levels of protective immunity. However, current methods to measure NAb are laboratory-based and are not necessarily suitable for large scale NAb monitoring in a large population. We develop a rapid test to detect SARS-CoV-2 NAb in 10 min that can be operated outside a laboratory. Our test provides results that are comparable to lab-based tests, which require between 1 h and up to 2 days to get a result. Our test may be useful for large-scale monitoring of immunity, for example in populations that do not have routine access to a lab.

[1] Antimicrobial Resistance Interdisciplinary Research Group (AMR-IRG), Singapore-MIT Alliance in Research and Technology (SMART), #03-10/11 Innovation Wing, 1 CREATE way, Singapore 138602, Singapore. [2] Department of Microbiology and Immunology, Yong Loo Lin School of Medicine, National University of Singapore (NUS), 5 Science Drive 2, Blk MD4, Level 3, Singapore 117545, Singapore. [3] School of Biological Science (SBS), Nanyang Technological University (NTU), 60 Nanyang Dr, Singapore 637551, Singapore. [4] Department of Medicine, National University Hospital (NUH), 5 Lower Kent Ridge Rd, Singapore 119074, Singapore. [5] The Infectious Diseases Translational Research Programme (ID TRP), NUS Yong Loo Lin School of Medicine, 1E Kent Ridge Road, Singapore 119228, Singapore. [6] Department of Chemical Engineering, Massachusetts Institute of Technology (MIT), 25 Ames Street, Building 66, Cambridge, MA 02139, USA. [7] Life Sciences Institute (LSI), National University of Singapore (NUS), Center for Life Sciences, #05-02, 28 Medical Drive, Singapore 117456, Singapore. [8] These authors contributed equally: Patthara Kongsuphol, Huan Jia, Hoi Lok Cheng, Yue Gu, Bhuvaneshwari D/O Shunmuganathan. [9] These authors jointly supervised this work: Paul A. MacAry, Hadley D. Sikes, Peter R. Preiser. ✉email: micpam@nus.edu.sg; sikes@mit.edu; prpreiser@ntu.edu.sg

COVID-19 affects > 200 million people and, to date, has killed > 4 million, worldwide. To prevent transmission of SARS-CoV-2—the virus that causes COVID-19—tight restrictions on movement and social interactions have been placed on populations across the globe. While this has had some effect on preventing the spread of the virus, they have plunged the global economy into a severe contraction. A phased relaxation of these social control measures is critical to allow business, and the world economy, to recover.

Achieving herd immunity against SARS-CoV-2, either naturally or through vaccination, is the ultimate long-term goal that will allow lifting of the widespread social control measures currently applied. Neutralizing antibodies (NAbs) are generated in response to either exposure to the virus, or to a vaccine. For effective prevention of viral infections, NAbs must be generated in sufficient quantity[1]. Screening populations for the presence of NAbs is essential to evaluate herd immunity against SARS-CoV-2, and to assess the effectiveness of vaccine immunization programmes, deployed in many countries since late 2020. To facilitate rapid screening of SARS-CoV-2 NAbs, NAbs detection tests that can be performed simply, rapidly and at low cost are highly desired.

Currently, NAbs are generally detected using virus neutralization tests (VNTs). Standard VNTs require handling of live virus (conventional VNT (cVNT)) or pseudovirus (pVNT), BSL3/BSL2 facility, skilled personnel, and 2–4 days processing time[2-5] making them unsuitable for mass testing the immune status of a population. SARS-CoV-2 initiates the process of host cell entry, by interacting with angiotensin-converting enzyme II (ACE2) receptors on the host cell via the receptor-binding domain (RBD) of the spike (S) protein[1,2,4,6-9]. Based on this observation, a rapid (1–2 h) plate-based ELISA, surrogate SARS-CoV-2 neutralization test (sVNT) has been developed using recombinant hACE2 receptor and viral RBD proteins[10]. NAbs are detected by their ability to bind RBD and block the formation of RBD/hACE2 complexes. Though much more rapid than the standard VNTs, the sVNT still require a laboratory setting and skilled personnel, presenting a barrier to large-scale screening.

Here we report a rapid cellulose pull-down viral neutralization test (cpVNT) that detects SARS-CoV-2 NAbs in plasma samples within 10 min and can be used at the point of care (POC). The test principle is based on the interaction of (i) RBD tagged cellulose-binding domain (CBD) (ii) ACE2 receptor tagged biotin (BA) and (iii) streptavidin conjugated horseradish peroxidase (SA-HRP), to detect NAbs binding to the RBD on cellulose paper. Despite the simplified and very rapid testing procedure, the cpVNT exhibits comparable performance to the lab-based tests in determining the level of NAbs in COVID-19 convalescent plasma samples with accuracy well above 80% and 90%, compared to pVNT and sVNT, respectively.

## Methods

**Materials.** Materials were purchased from the following sources, mouse anti-SARS-CoV-2 NAb (cat# 40591-MM43-100), monoclonal mouse anti Influenza A H10 Hemagglutinin/HA NAb (cat# 40359-M001), monoclonal mouse anti Influenza A Nucleoprotein IgG (cat# 11675-MM03T) and polyclonal rabbit anti-SARS-CoV-2 nucleocapsid protein IgG (cat# 40588-T62) from Sino Biological, USA; monoclonal rabbit anti MERS Coronavirus Spike protein NAb (cat# MA5-29975) and polyclonal rabbit anti-Dengue Virus Type 2 NS3 protein IgG (cat# PA5-32199) from Invitrogen, USA; polyclonal rabbit anti-Zika virus NS5 protein IgG (cat# GTX133312), polyclonal rabbit anti-Zika virus NS3 protein IgG (cat# GTX133309), monoclonal mouse anti-Dengue virus envelope protein IgG (cat# GTX629117) and

monoclonal mouse anti-SARS-CoV-2 spike protein IgG (cat# GTX632604) from GeneTex, USA. Other chemicals were of analytical grades from Merck, Singapore, otherwise stated.

**Collection of clinical samples.** The collection of COVID-19 convalescent samples were reviewed and approved under the DSRB reference # 2020/00120, National University of Singapore (NUS). Subjects with COVID-19 PCR positive results were recruited for this study. Informed consents including the use of clinical information were obtained from the participants for the COVID-19 convalescent samples. Peripheral blood was collected in EDTA blood tubes and subsequently diluted with an equal amount of sterile PBS. This was then gently layered on top of 13 mL Ficoll-Plaque density gradient media (GE Healthcare) in a 50 mL Falcon tube. The tube was centrifuged at 2400 rpm for 30 min with acceleration and deceleration set at 0. Plasma was harvested from the top layer and stored at −80 °C. Buffy coat layer was washed with sterile PBS at 2000 rpm for 6 min followed by another wash at 1500 rpm for 5 min. Peripheral blood mononuclear cells (PBMNCs) were harvested, resuspended in freezing media containing 90% FBS (Hyclone) + 10% dimethyl sulfoxide (Sigma Aldrich), and stored in liquid nitrogen.

Collection of pre-COVID samples were reviewed and approved by the Institutional Review Board of Nanyang Technological University, Singapore (IRB 003/2010, IRB 11/08/03, IRB 13/09/01, IRB-2016-01-045 and IRB-2020-11-047). The whole blood was donated by healthy adult volunteers at the National University Hospital, Singapore. Informed consents were obtained from all donors in accordance with the approved protocols. Whole blood samples collected were centrifuged at 2000 rpm for 10 min. Plasma was collected and stored at −80 °C until used.

**Isolation and cloning of SARS-CoV-2 Spike RBD-specific human antibodies.** Memory B cells were isolated from PBMNC derived from blood samples drawn from COVID-19 convalescent patients using a Human Memory B cell isolation kit (Miltenyi Biotec, #130-093-546). Small pools of purified Memory B cells were seeded into 384-well plates on irradiated CD40L-expressing feeder cells for differentiation into plasma cells as described previously[11]. After 7 days of culture, supernatants from B cell pools were screened for binding activity on SARS-CoV-2 Spike by ELISA. Antibody Heavy and Light Chain variable regions were cloned from positive wells by PCR (Collibri™ Stranded RNA Library Prep Kit for Illumina™ Systems) and whole human IgG reconstructed as described previously[12]. Confirmation of binding specificity of cloned human monoclonal antibodies was confirmed by ELISA.

**Protein expression and purification.** The soluble extracellular fragment of human ACE2 (residues 19–615; GenBank: AB046569.1) was cloned into a modified pHLSec[13] mammalian expression vector following an N-terminal monoFc, hexahistidine tag and Tobacco Etch Virus (TEV) protease cleavages site. SARS-CoV2-Spike RBD[14] fused to CBD (residues 276-434 of *Hungateiclostridium thermocellum* CipA) was cloned into the pHLmMBP-10 vector[15] (a gift from Luca Jovine; Addgene plasmid 72348) which encodes an N-terminal octahistidine tag, codon-optimized maltose-binding protein (MBP) tag and a TEV site. The coding sequence for the single-chain variable fragment (scFv) of the anti-SARS-CoV CR3022[16], and was subcloned into pHLmMBP-10 to generate an MBP-scFv fusion construct. Verified plasmids were transfected into Expi293F cells by using the Expifectamine293 transfection kit (ThermoFisher Scientific, #A1435101) to express the secreted proteins following the supplier's standard protocol. Cells were harvested by centrifugation

after 6 days of transfection and the supernatants were collected for protein purification. The media were conditioned for Ni-NTA binding by adding 2.5 mL of conditioning buffer, 200 mM HEPES pH 7.5, 3 M NaCl and 10% glycerol; 10 μL mammalian protease inhibitor cocktail (Nacalai Tesque, #25955-11) per 50 mL media. Proteins were first purified by affinity chromatography using Ni-NTA cartridges (Qiagen, #1046323), followed by size exclusion chromatography by using HiLoad 16/60 Sephadex 200 (Cytiva, formerly GE Healthcare) in gel filtration buffer (20 mM HEPES pH 7.5, 300 mM NaCl, 10% glycerol). To avoid protein cross-linking and aggregation, pooled fractions were supplemented with 0.5 mM TCEP before being concentrated by using Vivaspin centrifugal concentrators (Cytiva). The His-MBP tag of sRBD-CBD was cleaved off by using TEV protease (a gift of NTU Protein Production Platform, proteins.sbs.ntu.edu.sg) at 4 °C overnight with 1:40 mass ratio. Untagged sRBD-CBD was separated from His-tagged proteins by passing the reaction mixture through HisPur-Ni-NTA resin (Thermo Scientific, #88222) pre-equilibrated in 20 mM HEPES pH 7.5, 300 mM NaCl, 10 mM Imidazole. The purified sRBD-CBD sample was buffer exchanged and concentrated in 20 HEPES pH 7.5, 300 mM NaCl, 10% glycerol and 0.5 mM TCEP for storage.

SARS-CoV-2 N protein (residues 1–419; GenBank: YP_009724397.2) was synthesized by Genewiz (USA) and cloned into pET28b(+) bacterial expression vector following a hexahistidine tag and a thrombin cleavage site. Constructed plasmid was transformed into BL21 (DE3) competent cell for protein expression, briefly, culture in LB broth miller (1st Base, #BIO-4000-1kg) supplemented kanamycin (GOLDBIO, #K-120-25) was allowed to grow till $OD_{600}$ of 0.8 prior it was induced using IPTG (isopropyl β-D-1-thiogalactopyranoside) at final concentration of 0.5 mM for overnight at 16 °C. Bacterial cell pellet was then lysed in lysis buffer (20 mM Tris-HCl pH 7.9, 500 mM NaCl) supplemented with protease inhibitor cocktail (Nacalai Tesque, #04080-11) by sonication. Soluble portion was collected and incubated with HisPur-Ni-NTA resin for metal affinity purification. Size exclusion chromatography with HiLoad 16/60 Superdex 75 was carried out for final purification of SARS-CoV-2 N protein with gel filtration buffer (1 × PBS pH7.9). Collected protein fractions were pooled and concentrated with Vivaspin centrifugal concentrators prior storage at −80 °C.

**Biotinylation of monoFc-ACE2.** Chemical biotinylation of monoFc-ACE2 was carried out by using EZ-link Sulfo-NHS-LC-Biotinylation kit (ThermoFisher, #21435). Protein was incubated with 20 molar excess of Sulfo-NHS-LC biotin at 4 °C for 2 h. The level of biotinylation was measured by HABA assay provided from the kit.

**Antibody profiling by ELISA.** SARS-CoV-2 Spike protein, MBP-RBD protein, or nucleocapsid protein was coated on 96-well flat-bottom maxi-binding immunoplate (SPL Life Sciences, #32296) at 7.5 nM, 27 nM, or 40 nM, respectively, 100 μL/well at 4 °C overnight. The plate was washed three times in PBS and blocked for 2 h with blocking buffer: 4% skim milk in PBS with 0.05% Tween 20 (PBST) at 350 μL/well. After three washes in PBST, 100 μL of 80 times diluted plasma samples were added to each well for 1 h incubation. The plate was then washed three times in PBST and 100 μL of 5000 times diluted goat anti-human IgG-HRP (Invitrogen, #31413), or 5000 times diluted F(ab')2 anti-human IgA-HRP (Invitrogen, #A24458), or 7500 times diluted goat anti-human IgM-HRP (Invitrogen, #31415) was added to each well for 1 h incubation protected from light. After three times of plate wash in PBST, 100 μL of 1-Step Ultra TMB-ELISA (Thermo Scientific, #34029) was added to each well. After 3 min

incubation in dark, the reaction was stopped with 100 μL of 1 M $H_2SO_4$ and $OD_{450}$ was measured using a microplate reader (Tecan Sunrise). $OD_{450}$ reported was calculated by subtracting the background signal from plasma binding to the blocking buffer.

**Bio-layer interferometry (BLI).** The N-terminally biotinylated monoFc-ACE2 interaction with RBD–CBD was measured on an 8-channel Octet RED96e system (Forté Bio) with streptavidin biosensor tips (Sartorius). These tips were pre-incubated with assay buffer: PBS, 0.2% BSA and 0.05% Tween 20 for 10 min at 25 °C. Then, they were coated with biotinylated mFc-ACE2 to yield a loading thickness of 0.9 nm. After washing the tips with assay buffer, the binding with RBD–CBD was measured in real time by recording the increase in optical thickness of the tips during 600 s of the association phase. The tips were transferred back into assay buffer during the dissociation phase. A two-fold dilution series of RBD–CBD ranging from 6.25 to 100 nM was used. For negative control, the concentration of N-protein was kept at 100 nM for comparison with the highest concentration of RBD–CBD. The data were processed by Octet Data Analysis software then transferred into GraphPad Prism 9 for association-dissociation non-linear regression model curve-fitting.

**SARS-CoV2 pseudotyped lentivirus production.** This method was optimized from Poh et al.[17] A third-generation lentivirus system, was used to produce pseuduotyped viral particles expressing SARS-CoV2 S proteins via reverse transfection. $36 \times 10^6$ HEK293T cells were transfected with 27 μg pMDLg/pRRE (Addgene, #12251), 13.5 μg pRSV-Rev (Addgene, #12253), 27 μg pTT5LnX-WHCoV-St19 (SARS-CoV2 Spike) and 54 μg pHIV-Luc-ZsGreen (Addgene, #39196) using Lipofectamine 3000 transfection reagent (Invitrogen, #L3000-150) and cultured in a 37 °C, 5% $CO_2$ incubator for 3 days. The viral supernatant was then, harvested and filtered through a 0.45 μm filter unit (Merck). The filtered pseudovirus supernatant was concentrated using 40% PEG 6000 by centrifugation at 1600g for 60 min at 4 °C. Lenti-X p24 rapid titer kit (Takara Bio, #632200) was used to quantify the viral titres, as per the manufacturer's protocol.

**Pseudovirus neutralization assay (pVNT).** This method was modified from Poh et al.[17] The ACE2 stably expressed CHO cells were seeded at a density of $5 \times 10^4$ cells in 100 μL of complete medium [DMEM/high glucose with sodium pyruvate (Gibco, #10569010), supplemented with 10% FBS (Hyclone, # SV301160.03),10% MEM Non-essential amino acids (Gibco, #1110050), 10% geneticin (Gibco, #10131035) and 10% penicillin/streptomycin (Gibco, #15400054)] in 96-well white flat-clear bottom plates (Corning, #353377). Cells were cultured at 37 °C with the humidified atmosphere at 5% $CO_2$ for one day. Patient plasma samples were diluted to a final dilution factor of 80 with PBS. The pseudovirus is diluted to a final concentration of $2 \times 10^6$ PFU/ ml. In 25 μl there will be 50,000 lentiviral particles. The diluted samples were incubated with an equal volume of pseudovirus to achieve a total volume of 50 μL, at 37 °C for 1 h. The pseudovirus-plasma mixture was added to the CHO-ACE2 monolayer cells and left incubated for 1 h to allow pseudotyped viral infection. Subsequently, 150 μL of complete medium was added to each well for further incubation of 48 h. The cells were washed twice with sterile PBS. 100 μL of ONE-glo™ EX luciferase assay reagent (Promega, #E8130) was added to each well and the luminescence values were read on the Tecan Spark

100 M. The percentage neutralization was calculated as follows:

$$\text{Neutralization \%} = \frac{\text{Readout (unknown)} - \text{Readout (infected control)}}{\text{Readout (uninfected control)} - \text{Readout (infected control)}} \times 100\%$$

**Modified ELISA-based sVNT**. ACE2-Fc was conjugated to peroxidase using Peroxidase Labeling Kit- $NH_2$ (Abnova, #KA0014) according to the manufacturer's protocol. Each well of 96-well flat-bottom maxi-binding immunoplate was coated with 100 μL of 13 nM MBP-RBD at 4 °C overnight. The plate was washed and blocked as described above. The plate was washed three times in PBST and incubated for 1 h with 100 μL/well of plasma samples diluted ten times in blocking buffer. No inhibitor control wells were incubated with blocking buffer. Positive and negative control wells were established by incubating with functionally characterized recombinant monoclonal antibodies targeting SARS-CoV-2 RBD. A characterized neutralizer was included as the positive control and a non-neutralizing binder was included as the negative control. Both monoclonal antibodies were tested at concentrations from 64 nM to 0.5 nM, prepared via 2× serial dilution in the blocking buffer. Subsequently, the plate was washed three times and incubated for 1 h with 0.4 nM ACE2-Fc-peroxidase, 100 μL/well, protected from light. The following steps of color development and absorbance measurement were performed as described above. Inhibition% was calculated as

$$\text{Inhibition \%} = \left( \frac{OD450 \text{ of negative control (no inhibition)} - OD450 \text{ of sample}}{OD450 \text{ of negative control (no inhibition)}} \right) \times 100\%$$

**cpVNT assay**. Cellulose test strips were prepared using Whatman No. 1 chromatography paper (GE healthcare, #3001-861). The papers were printed with wax-ink printer (Xerox ColorQube 8570, Xerox, USA) to define liquid flow path and testing region. The non-testing regions were printed with the wax ink whereas the testing region were left unprinted. Circular testing region with diameters of 5 mm and 6 mm were prepared. The printed papers were baked at 150 °C for 1 min to allow the wax ink to diffuse through the paper forming hydrophobic boundary throughout the paper thickness. The wax-free testing regions were blocked with 10 μL of 5% BSA in PBS. After air-drying, the test strips were stacking into three layers with the 5 mm strips on the topmost layers and 6 mm strips on the second and third layers. The three layered wax printed paper allows consistent flow of liquid at ~10 s when 40 μL of liquid are applied. One piece of Kimwipes paper (11.4 cm × 21.6 cm, Kimberly-Clark Professional, # 34155) folded in half for 6 times was used as absorbent pad. The three-layered test strips were stacked on top of the folded Kimwipes. All layers were secured together using two paper binders.

10 nM RBD–CBD in 1% BSA in PBS was prepared and assigned as reagent "A". 10 nM biotinylated monoFc-ACE2 with 6 nM SA-HRP (Biolegend, #405210) in 1% BSA in PBS was prepared and assigned as reagent "B". To perform the test, one part of reagent A and one part of reagent B were mixed with two parts of plasma samples, i.e. for one reaction, the mixture contains 10 μL of A, 10 μL of B and 20 μL of sample. The mixture was incubated for 5 min at room temperature. 40 μL of the mixture was applied to the testing region. Once sample was fully absorbed the test was washed once with 40 μL of PBS, followed by 40 μL of $TMB/H_2O_2$ solution (Merck, #T0440). Signals were allowed to develop for 3 min. Images were taken using Xiaomi Redmi A9 phone in a light box equipped with LED lights and save as.jpg format. Images were transferred to a PC. and analyzed using the opened source ImageJ software from NIH. Images were

converted from RGB color space to CMYK. Cyan intensity in the testing regions were analyzed. Inhibition% was calculated using the following formula:

$$\text{Inhibition \%} = \left( 1 - \frac{\text{Cyan intensity of sample}}{\text{Cyan intensity of negative control (no inhibition)}} \right) \times 100\%$$

**Pearson's correlation**. Pearson's correlation coefficiency was calculated using Microsoft Excel function PEARSON.

**Calculation of test performance**. Disease prevalence was calculated from the sample size. It may not represent the true prevalence. Calculations of each parameter of test performance were done using the following formula:

$$\text{Sensitivity} = \frac{\text{True positive}}{\text{True positive} + \text{False Negative}}$$

$$\text{Specificity} = \frac{\text{True negative}}{\text{False positive} + \text{True negative}}$$

Positive predictive value(PPV)

$$= \frac{\text{Sensitivity} * \text{prevalence}}{((\text{Sensitivity} * \text{prevalence})) + ((1 - \text{Specivicity}) * (1 - \text{prevalence}))}$$

Negative predictive value(NPV)

$$= \frac{\text{Specificity} * (1 - \text{prevalence})}{((1 - \text{sensitivity}) * \text{prevalence})) + ((\text{specificity} * (1 - \text{prevalence})))}$$

Accuracy = (Sensitivity * Prevalence) * (Specificity * (1 − Prevalence))

**Statistics and reproducibility**. All data points were performed at least in triplicates. Each data point represented a mean value with an error bar that represented standard deviation (SD). Some data points from clinical samples that were grouped together may not represent error bar. These data points represent mean value from at least three separate run.

**Reporting summary**. Further information on research design is available in the Nature Research Reporting Summary linked to this article.

## Results

**Optimization of rapid paper-based cpVNT**. The assay time for the currently established lab-based NAb tests range from 1.5 h to 4 days. A shortening of the overall assay time and a simplified workflow are primary requirements for POC NAb tests suitable for large-scale surveillance applications. Vertical-flow assay format allows reagents to flow in a top-to-bottom fashion via a short liquid flow path for the detection of biomolecules. This feature enables rapid and controllable flow speed for handling of reagents[18–21]. With an aim for a rapid POC NAb test, the vertical-flow assay format was selected for this study. Here we demonstrate that cellulose can be used as a test matrix for vertical-flow assays. The assay reaction is immobilized on the cellulose matrix via high-affinity interactions between the CBD and the cellulose matrix[21,22] by fusing CBD to the capture reagent[21–24]. Adopting cellulose as the test material bypasses the surge of high nitrocellulose demand from the global ramp up of rapid COVID-19 tests which represent a massive risk onto the supply chain.

The test principle relies on a complex formation between RBD/ACE2 receptor whereby the presence of NAb interferes with the RBD/ACE2 receptor complex thereby reducing the reporting

signal intensity. To enable the test to be compatible to cellulose paper, CBD was tagged to RBD (RBD–CBD) allowing the RBD to be captured rapidly and at high affinity onto cellulose surface[23]. ACE2 receptor was engineered to be a reporting molecule. This was done by tagging biotin (BA) onto ACE2, creating ACE2-BA. Horse radish peroxidase (HRP) conjugated streptavidin (SA), SA-HRP, was used as a colorimetric signal generator. A complex of ACE2-BA/SA-HRP was used to generate colorimetric signal via application of 3,3′,5,5′-tetramethylbenzidine (TMB)/$H_2O_2$, in which HRP catalyzes oxidation of TMB substrate, producing blue color signals.

To construct the test, recombinant (i) RBD–CBD and (ii) biotin (BA) tagged monoFc-ACE2 receptor proteins were expressed, purified (Supplementary Fig. 1a) and evaluated for their kinetic properties using BLI. Various concentrations of RBD–CBD ranging from 6.25 to 100 nM were used. Data showed specific binding between monoFc-ACE2 and RBD–CBD even at a low concentration of RBD–CBD at 6.25 nM (Supplementary Fig. 2b). The pair demonstrated high affinity toward each other with a $K_D$ of 12.7 nM (Supplementary Fig. 1b) which is comparable to previously reported $K_D$ ranging from 4.7 to 15.2 nM[25–28]. In contrast, SARS-CoV-2 structural, nucleocapsid (N) protein was tested on ACE2-BA to assess specificity of the protein. BLI data showed minimal interaction of BA-ACE2 with N protein even at 100 nM of N (Supplementary Fig. 1c), indicating that BA-ACE2 is highly specific to RBD.

To engineer the vertical-flow assay (Fig. 1), wax ink was printed onto cellulose paper to create hydrophobic boundary and define liquid flow path. Liquid flow rate was controlled by stacking three layers of cellulose paper[21] (Fig. 1a, d). The top layer has a 5 mm diameter of hydrophilic area without wax (testing region) and the lower two layers have 6 mm diameters of the hydrophilic areas. One unit of cellulose test strip comprises two testing regions (Fig. 1a, b). The distance between the center of the two testing spots is 12.5 mm. (Fig. 1a, b, d). A folded Kim Wipes paper was used as the absorbent pad (Fig. 1a, e). A 2 mm thick acrylic sheet was cut using a laser cutter (Epilog Fusion Edge Laser System, USA) into two pieces of an acrylic manifold, each with a dimension of $30 \times 50$ mm² (Fig. 1a–c, f). One of the pieces contains an opening of $25 \times 10$ mm² (Fig. 1a, c). Another piece of acrylic carries two pieces of 3 mm thickness acrylic sheets which were used to form spacers (Fig. 1a, b, f). Three-layered cellulose papers were stacked on top of the folded Kim Wipes and secured together using the acrylic manifold and paper binders (Fig. 1a, b). The spacers between the two pieces of the manifold provide consistent pressure between different cellulose test units. The test strip units provided a consistent liquid flow speed of ~10 s when 40 µL of liquid were used.

To capture the colorimetric signal from the cellulose vertical-flow assay, Xiaomi Redmi A9 phone was used to capture the image and save the image in a.jpg format. To fix the camera distance and angle as well as to prevent interference from the surrounding light, a 'light box' was created. The box has a $W \times L \times H$ dimension of $150 \times 230 \times 90$ mm³ (Fig. 1g). A void was made at the top face of the box to provide an access for the phone camera (Fig. 1h). The distance between the cellulose test unit to the camera was 85 mm. Fixtures were made to fix the locations of the test strip unit and the phone. The test strip unit fixture was secured to the base of the light box (Fig. 1i) and the camera fixture was secured to the top face of the box (Fig. 1g, h). Internal faces of the box were equipped with warm-white LED light strips (Fig. 1I). To prevent shadows on the cellulose test unit, white,

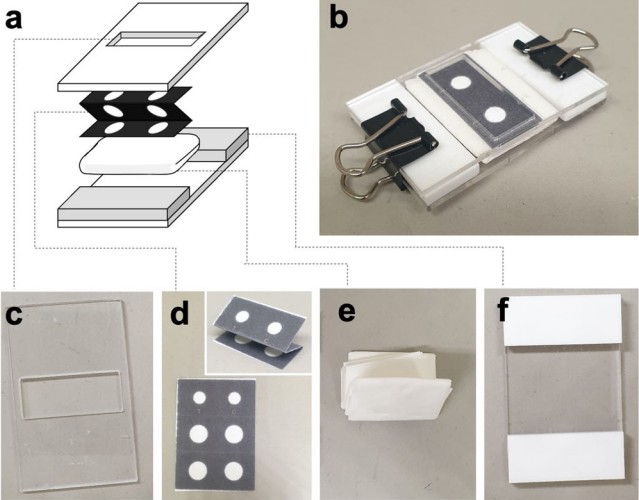
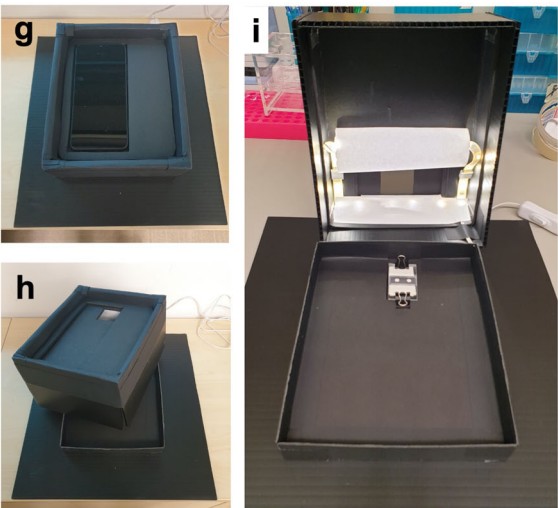

**Fig. 1 Overview of the test construction and image acquiring devices. a** Exploded scheme of the cellulose testing unit, comprising 2 pieces of acrylic manifold to hold the testing unit together, 3 layers of the folded and printed cellulose test strip whereby hydrophobic ink (black area) was used to determine liquid flow path and folded Kim Wipes which is used as an absorbent pad. **b** An Image of the assembled cellulose testing unit which is held together using paper binders. **c** An image the top piece of the acrylic manifold which contains an opening to access the cellulose test strip. **d** An image of cellulose test strip. The strip was printed and cut as one piece. Wax ink was used for printing (black area) to define the liquid flow path. To create 3 layers of the test strip, the printed paper is folded in a zig-zag motion until the hydrophobic regions are aligned (inset). The top most layer of the cellulose paper contains a circular testing region (white, hydrophilic area without ink) with a diameter of 5 mm whereas the lower two layers, each, contain hydrophilic regions with a diameter of 6 mm. Each cellulose testing unit contains two test zones for testing of two reactions. **e** An image of the folded Kim Wipes which was folded in half for 6 times and used as absorbent material. **f** An image of the lower piece of the acrylic manifold which contain 3 mm spacers at two ends. The spacers are used to control consistent pressure that applies to different cellulose testing units. **g** An image of the light box with the phone placed at the top face. **h** An image of a slight opened light box showing phone fixture and a void for phone camera access at the top face. **i** An image of opened light box showing internal structure of the box. Warm-white LED light strips were attached to the inner top face of the light box to provide consistent lighting for all images. Cylindrical, white papers were covered parts of the LED strips to diffuse light from the strip, prevent shadow which may form on the cellulose test unit. Fixture was placed at the base of the light box to provide a fixed location to place the cellulose test unit.

cylindrical shaped papers were used to cover the LED strip, diffusing light from the LED strips. Example of images captured from the phone camera and the light box are shown in Supplementary Fig. 1d. Images were analyzed using the open-source ImageJ software from National Institute of Health (NIH), USA. Only the circular hydrophilic testing regions were used for analysis. To optimize the image analysis, images were separated into different channels in RGB and CYMK color spaces. HRP/TMB signals generated on different cellulose papers were compared. Signals recorded from cyan channel provided the highest signal (RBD–CBD + ACE2-BA/SA-HRP + TMB) over noise (ACE2-BA/SA-HRP + TMB) ratio as compared to other channels in the RGB and CYMK color spaces. Therefore, cyan intensity in the CYMK color space was used for image analysis.

Different cellulose vertical-flow assay formats were performed to determine the most effective assay workflows (Supplementary Fig. 2). Results showed that application of the reporting complex (ACE2-BA/SA-HRP) onto the pre-immobilized RBD–CBD (group (iv) in Supplementary Fig. 2) generated only a slight change in cyan intensity as compared to control groups (group (i)–(iii) in Supplementary Fig. 2). We speculated that the rapid 10 s. liquid flow speed led to the inefficient capture of the reporting complex onto the RBD–CBD and cellulose paper. Substantial signal improvement was observed when the reaction was entrapped on the paper for 5 min (group (v) in Supplementary Fig. 2). However, complicate assay workflow would be required to maintain the assay reaction onto the cellulose surface, therefore this design is not ideal for POC settings. CBD is known to interact rapidly and effectively with cellulose matrix. Based on this knowledge, we optimized the assay by mixing all reagents in one 'pre-mix' solution and incubated for 5 min prior to applying the reaction to the cellulose test unit. With this design, the cellulose test unit produced highest cyan signal intensity as compared to other designs (group (vi) in Supplementary Fig. 2). Based on this optimization, the 'pre-mix' format is selected for the rapid NAb test and referred to as cellulose pull-down VTN (cpVNT) following the assay principle of the optimized test format. In addition, protein kinetic data from BLI experiment showed that >80% of RBD/ACE2 receptor complex can be formed within 5 min (Supplementary Fig. 1b), ensuring that most of the RBD/ACE2 complexes were formed within the incubating period.

Based on these optimizations, the assay can be performed by mixing the plasma sample with the reagents and incubating for 5 min to allow the efficient formation of RBD–CBD/NAbs or RBD–CBD/ACE2 complexes in aqueous phase before applying it onto the cellulose paper (Fig. 2a). The vertical-flow assay format ensures that most of the RBD–CBD complexes formed are exposed to the paper matrix and effectively captured through the rapid and high-affinity interaction of CBD and cellulose. The high affinity of CBD to cellulose ensures a minimal loss of the complex during the washing step.

Concentrations of RBD–CBD and ACE2-BA/SA-HRP were further optimized to obtain the highest signal difference between the presence and absence of NAb (Supplementary Fig. 3). To do so, human serum containing 0 or 100 nM NAb were used for signal comparison. The ratio between maximal and minimal cyan intensities obtained from 0 and 100 nM NAb, respectively, were used to calculate the signal ratio. The concentration of ACE2-BA was first optimized using fixed concentrations of 20 nM and 2 nM of RBD–CBD and SA-HRP, respectively. Signals were captured at 3 min following the addition of TMB and washing solution. Results showed that 10 nM ACE2-BA produced the highest cyan intensity ratio from 0 nM:100 nM NAb (Supplementary Fig. 3a, b). Therefore 10 nM ACE2-BA was selected for subsequent experiments. To optimize for RBD–CBD concentration, ACE2-

BA and SA-HRP concentrations were fixed at 10 nM and 2 nM, respectively. Two different concentrations of RBD–CBD, 10 and 20 nM were tested. Results showed that both concentrations produced a similar cyan intensity ratio from 0 nM:100 nM NAb which were at ~2.1. Therefore, different NAb concentrations were tested to further explore the different cyan intensity signals at various NAb concentrations. This showed that RBD–CBD at 10 nM produced distinguishable cyan intensity signals at lower NAb concentrations (Supplementary Fig. 3c, d). This concentration was selected for further studies. For optimization of SA-HRP, RBD–CBD and ACE2-BA concentrations were fixed at their optimized concentrations of 10 nM. Various concentrations of SA-HRP were tested in serum containing 0 or 100 nM NAb. Results showed that 6 nM SA-HRP produced the highest ratio of cyan intensity from 0 nM: 100 nM NAb (Supplementary Fig. 3e, f). Therefore 6 nM SA-HRP was selected for subsequent analysis. Altogether, optimized concentrations of RBD–CBD, ACE2-BA and SA-HRP for cpVNT were 10 nM, 10 nM and 6 nM, respectively.

**The rapid paper-based cpVNT performance**. To validate the cpVNT test, different concentrations of SARS-CoV-2 NAbs were spiked in non-diluted human plasma and evaluated for their ability to inhibit RBD–CBD/ACE2 complex formation (Fig. 2a). This approach demonstrated efficient inhibition of RBD–CBD/ACE2 complex formation for a dynamic range of 10–100 nM (Fig. 2b, c). We tested the cpVNT using plasma samples containing non-neutralizing human anti-RBD antibodies produced from our lab. Results showed that only minimal inhibitory signals were observed from these samples even at high concentrations of the non-NAbs (Fig. 2b, c). These data indicate that the test is highly specific to the SARS-CoV-2 NAbs but not to non-NAbs. The data show a strong relationship between antibody concentrations and inhibition of complex formation (Fig. 2c) with a limit of detection (LOD), calculated by using 'mean negative + 3 SD' formula, of 10 nM NAbs.

**Assessments of SARS-CoV-2 immunological profile from COVID-19 convalescent plasma**. Prior to validating the cpVNT, general assessments of the clinical samples were performed. Plasma samples from 24 confirmed COVID-19 patients were collected between 29 days and 73 days (median of 49 days) post positive PCR test (follow up visit 1 (FV1)). Subgroups of patient plasma samples were collected on two subsequent occasions (FV2 ($n = 13$); and FV3 ($n = 10$)) (Supplementary Table 1). General assessments were performed, for each convalescent plasma sample, to obtain immunological profiles against SARS-CoV-2 and to determine reference points for cpVNT evaluation. Antibody subtypes IgA, IgM and IgG against SARS-CoV-2's S, RBD, and nucleocapsid (N) proteins were evaluated using ELISA (Supplementary Fig. 4). IgA and IgM levels against RBD peaked in FV1 in several samples and decline to baseline in FV2 and FV3. IgA and IgM against S and N were found to be low throughout the enrollment period. The minimal levels of IgA and IgM observed in this study are in good agreement with recent studies[29,30] showing that IgA and IgM against RBD and N peaked at 14–20 days post symptom onset and started to wane after ~20 days. Plasma samples collected for this study began on day 29 post-admission in which IgA and IgM are expected to be low or declining. IgG levels were found to be higher than IgA and IgM (Supplementary Fig. 4). No statistical difference in IgG levels against all three SARS-CoV-2 biomarkers were observed from FV1 and FV2 (Supplementary Fig. 5). A significant reduction of 15%–25% in IgG levels against all biomarkers were seen in FV3 as compared to FV2 (Supplementary Fig. 5). These findings suggest

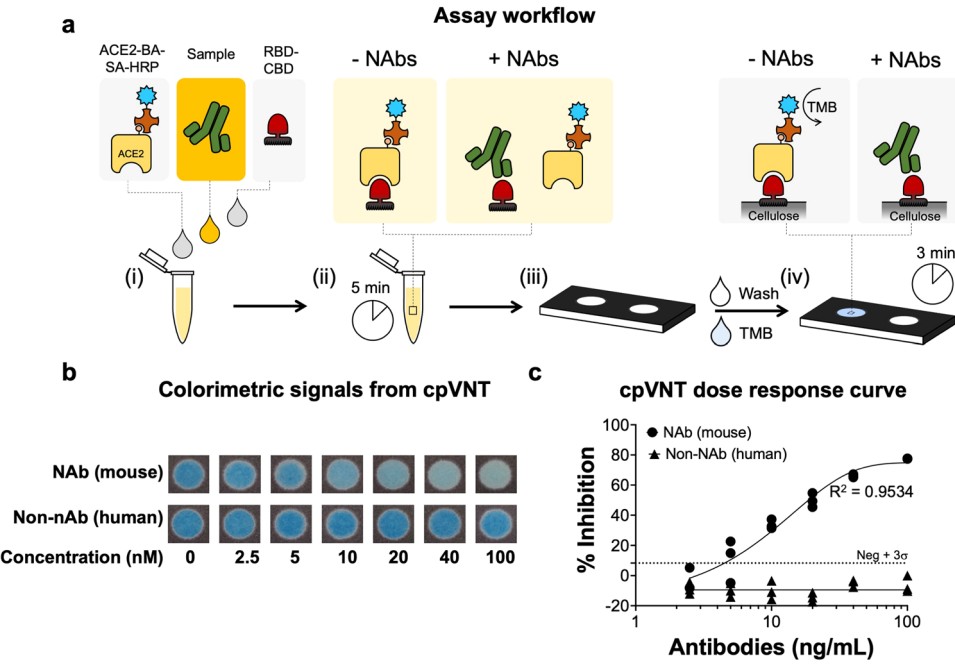

**Fig. 2 Overview of workflow and signal analysis of cellulose pull-down VNT. a** Schematic of working principle and workflow of cellulose pull-down VNT (cpVNT). (i) Un-diluted plasma is mixed with receptor binding domain (RBD) tagged to cellulose binding domain (RBD–CBD) and angiotensin-converting enzyme 2 receptor (ACE2) tagged to biotin and streptavidin horse-radish peroxidase (ACE2-BA-SA-HRP). (ii) The reaction is incubated for 5 min to allow neutralizing antibodies (Nabs)/RBD-CBD or ACE2-BA-SA-HRP/RBD–CBD complex formations. (iii) The mixture is applied on the cellulose-based vertical-flow device. (iv) Washing solution and ready-to-use 3,3',5,5'-tetramethylbenzidine (TMB) substrate solution are sequentially applied to the device. Colorimetric reaction is allowed to develop for 3 min. Signals were captured using camera. **b** Images of cpVNT test results at different concentrations of mouse anti-SARS-CoV-2 neutralizing antibodies and human anti-RBD non-neutralizing antibodies. **c** Plots of inhibitory percentages derived from cyan intensity signals. Limit of detections (LOD) were determined using mean + 3 standard deviation (SD) formula and represented as the dotted lines. All data were represented as mean ± SD. Each data points were performed in triplicates. 4 Parameters logistic model was used to draw the fitted curve with $R^2$ value of 0.9534.

that IgG levels against S, RBD and N proteins were maintained at a high level for around 4 months post-admission and started to decline slowly after 5–6 month.

To determine the levels of NAbs present in different plasma samples, two lab-based VNTs were used (i) chemiluminescent-based pseudovirus VNT (pVNT) and (ii) a modified format of the published ELISA-based surrogate VNT (sVNT)[10]. pVNT provides results that are closely correlated to conventional virus neutralization test using live virus[31,32]. Therefore it is widely used as a substitute method for conventional Plaque Reduction Neutralization Test (PRNT) and in vitro live virus neutralization assays, to quantitively assess NAb status[17,33–37]. The established pVNT protocols[17] was used as a reference for pVNT performed in this study. pVNT determined NAb status by measuring chemiluminescent signals from cells infected with pseudovirus. The presence of NAbs prevents the virus from infecting the cells thereby reducing chemiluminescent signals. Based on the test optimization and the reference study[17], a wide range of sample dilution factors from $10^1$–$10^6$ were used to determine the effective cut-off between the presence and absence of NAb. A 50% signal inhibition was indicated as an effective value to distinguish between the presence and absence of NAb. Similarly, pVNT established by different research groups also demonstrate 50% signal inhibition as an effective cut-off. During the test optimization, we found that a dilution factor of at least 1:80 is necessary to minimize false positives produced by healthy control samples, therefore a fix 1:80 dilution factor is used in this study as a simplified pVNT. The chemiluminescent signals measured were normalized with the signals from non-infected and pseudovirus

infected cells. The status of NAbs was expressed as neutralization percentage.

For sVNT, while it retains the same test principles as the published method[10], the modified sVNT configuration used recombinant RBD protein as a capture reagent and ACE2 receptor conjugated with HRP as a reporter reagent. In this test format, the presence of NAbs causes signal reduction that can be expressed as inhibitory percentages. The higher percentages represent a higher level of NAbs and vice versa. 10× dilution were chosen as the lowest dilution factors that give reliable results without showing false positives from healthy control samples. The cut-off value for sVNT was determined by analyzing the test sensitivity and specificity using known concentrations of NAbs and non-NAbs spiked in plasma samples (Supplementary Fig. 6). Receive operating characteristic (ROC) curve showed that the test maintained a high sensitivity of 75% while achieving a high specificity of 100% at the inhibitory percentage of ~20%, therefore 20% inhibition is used as a cut-off value to distinguish positive from negative NAb status. Using this cut-off, the sVNT is sensitive to detect RBD-specific NAbs at a concentration of 313 ng/mL (2 nM), much lower than the average concentration of these antibodies in COVID-19 convalescent patients[38].

Data obtained from pVNT and sVNT are shown in Fig. 3a, b, d, e. The pVNT and sVNT exhibit a high Pearson correlation coefficient of 0.8 (Fig. 3c, f), indicating that both tests produced results that are in good concordance. In addition, we observed that NAb status is highly correlated to IgG levels against S and RBD with Pearson coefficients of 0.72 and 0.74, respectively (Supplementary Fig. 7a, b) and moderately correlated with IgG level against N with a Pearson coefficient of 0.59 (Supplementary

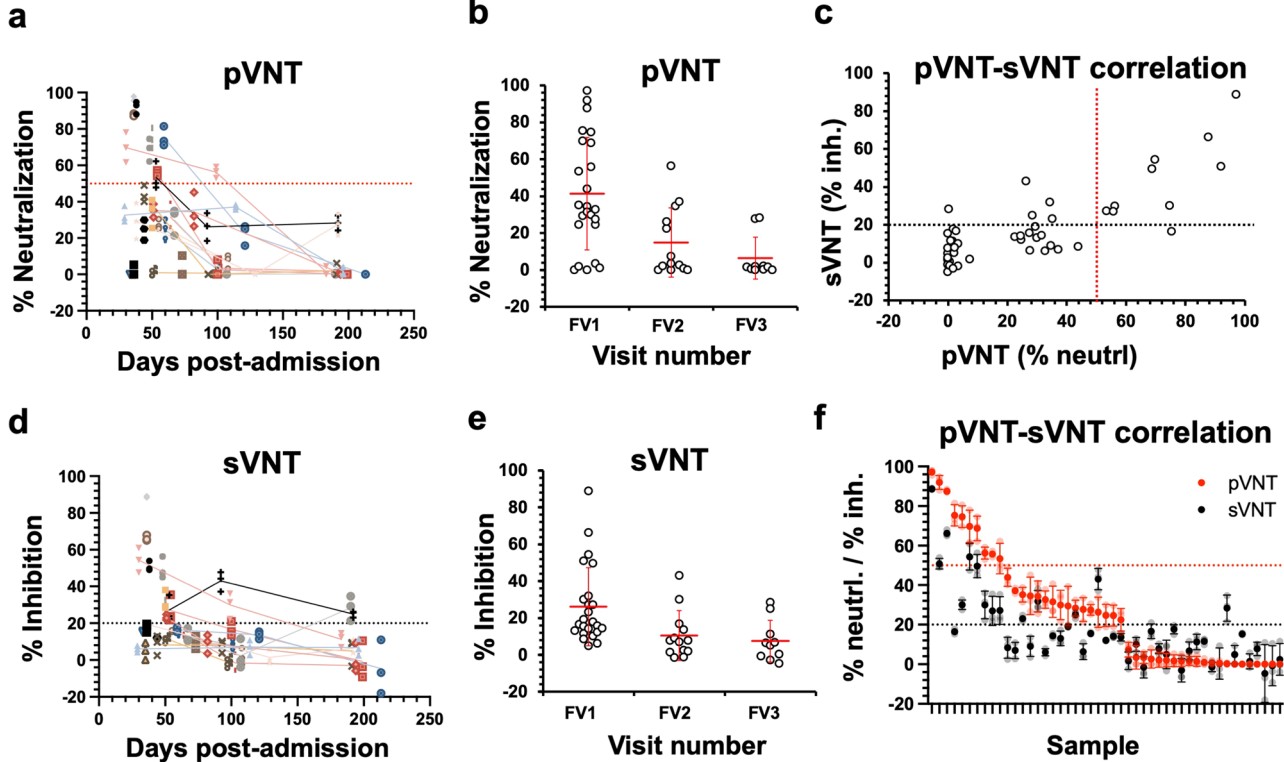

**Fig. 3 Comparison of COVID-19 neutralizing antibodies (NAb) status from convalescents samples between pseudovirus neutralization test (pVNT) and surrogate virus neutralization test (sVNT) methods.** Evaluation of NAb status from COVID-19 convalescent samples at different time points post infection from different follow up visits (FV) using (**a**, **b**) pVNT and (**d**, **e**) sVNT. Data shown in (**a**, **d**) were mean ± standard deviation (SD) of % neutralization for pVNT and mean ± SD% inhibition for sVNT, respectively. Each data points were performed in triplicates. Lines connecting different data points represented samples that were obtained from the same subjected collected at different time points. Data shown in (**b**, **e**) were mean values of % neutralization for pVNT and % inhibition for sVNT from different samples. **c** Correlation between sVNT and pVNT. Mean values from each data point were represented on the graph. **f** Alternative correlation plot between sVNT and pVNT where data were arranged from highest to lowest values of pVNT neutralization percentages. Data were represented as mean ± SD. Red and black dotted lines represent cut-off values at 50% and 20% for pVNT and sVNT, respectively, to distinguish positive and negative NAb status.

Fig. 7c). No clear correlation is observed between NAb status and disease severity in our study (Supplementary Fig. 7d), in line with the recent observation reported by Röltgen et al.[39].

pVNT presents a test format that is closely related to events that occur in a physiological condition. Therefore, pVNT will be used as a baseline to determine the accuracy of sVNT. Using the defined 50% cut-off for pVNT and 20% cut-off for sVNT, the sVNT provides test sensitivity and specificity of 90.0% and 86.5%, respectively with an overall accuracy of 87.2% (with 95% CI of 74.3% to 95.2%) (Supplementary Table 2A). It is also observed that most samples showed negative NAb status (Fig. 3). Majority of the positive NAbs detected came from FV1 samples in both test formats.

**Evaluating of cpVNT performance using COVID-19 convalescent plasma.** To evaluate the ability of the cpVNT in determining the level of NAbs, results obtaining from cpVNT using COVID-19 convalescent plasma, from different visits, were plotted against the pre-COVID plasma samples collected from earlier studies. Based on these data, the inhibitory signal cut-off which distinguished positive from negative NAbs levels was determined at 20% (Fig. 4a). Negative inhibitory percentages were observed from some data points. The negative values were due to the higher cyan intensity signals as compared to the reference point calculated from the mean value of pre-COVID and non-infected samples. In the context of cpVNT, it can be interpreted that no NAbs were detected from these samples. In a similar

fashion to pVNT and sVNT (Fig. 3), results from cpVNT showed that most of the COVID-19 convalescent samples exhibit negative NAb status with positive status observed mostly from FV1 (Fig. 4b, c). An average value of NAbs from FV1 falls between the designated cut-off value and average values from FV2 and 3 fall below the cut-off value. Similar trends were also observed from pVNT and sVNT (Fig. 3b, e).

Data obtained from cpVNT show a high correlation with pVNT and sVNT with Pearson correlation coefficients of 0.70 and 0.87, respectively, (Fig. 4e, e and Supplementary Fig. 8a, b). As compared to pVNT, cpVNT exhibits the test sensitivity and specificity of 80.0% and 84.4%, respectively, with an overall test accuracy of 83.3% (with 95% CI of 68.6% to 93.0%, Supplementary Table 2B). As compared to sVNT, cpVNT exhibits a sensitivity and specificity of 85.71% and 96.55%, respectively, with an overall test accuracy of 93.02% (with 95% CI of 74.37% to 96.02%, Supplementary Table 2c).

Cross reactivity tests were performed to ensure the test specificity. High concentrations (100 nM) of IgG against different viruses were spiked in plasma from healthy controls and tested on cpVNT. Data show minimal cross reactivity to antibodies against other viruses, or to SARS-CoV-2 S and N proteins (Fig. 4f), which possess non-neutralizing ability, suggesting that cpVNT is highly specific to SARS-CoV-2 NAbs. cpVNT specificity is comparable to the commercialized lab-based sVNT[40]. In addition to plasma, we also tested the cpVNT using human serum samples. Results showed dose–response curves that are comparable to the plasma

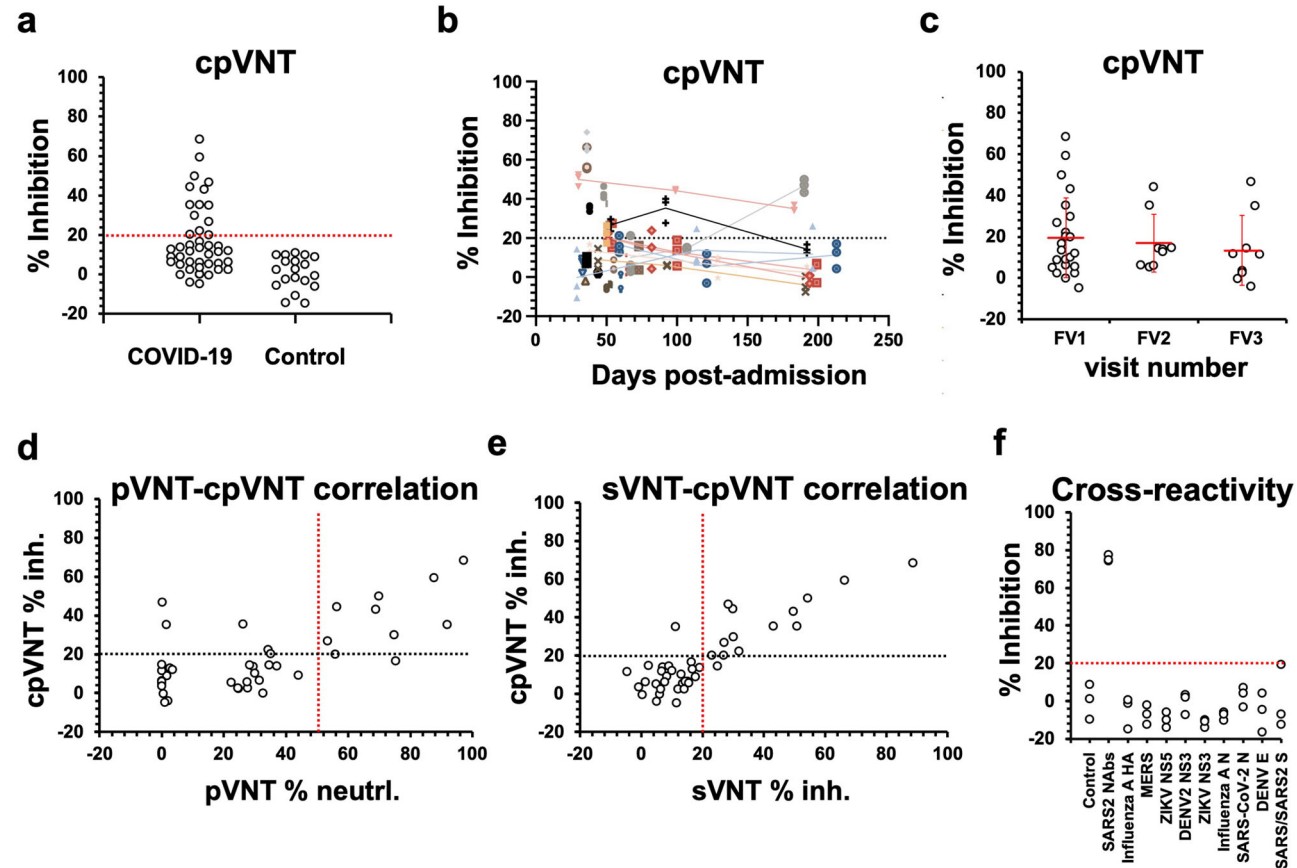

**Fig. 4 Evaluation of neutralizing antibodies (NAb) status from COVID-19 convalescent samples, at different time points post infection, from different follow up visits (FV) using cellulose pull-down virus neutralization test (cpVNT). a** NAbs from all samples and visits as compared to pre-COVID plasma samples. Cut-off value which distinguished positive from negative NAb levels was designated at 20% inhibitory percentage (red line). Each data point represented mean value of % inhibition of different samples. Each point was performed in triplicates. **b** NAbs detection using cpVNP from different samples at different time points. Data were represented as mean ± standard deviation (SD). Each data point was performed in triplicates. Lines connecting different data points represented signals obtained from the same sample at different time points. **c** Alternative presentation of NAbs detected from different visits. Each data point represented mean value of % inhibition obtained from different samples. Each point was performed in triplicates. **d** Correlation between cpVNT and pseudovirus neutralization test (pVNT). Black and red lines represent cut-off values at 20% and 50% for cpVNT and pVNT, respectively. Each data point represented mean value of % inhibition or % neutralization for cpVNT and pVNT, respectively. Each point was performed in triplicates. **e** Correlation between cpVNT and surrogate virus neutralization test (sVNT). Black and red lines represent cut-off values at 20% for cpVNT and pVNT. Each data point represented mean value of % inhibitions for cpVNT and pVNT. Each point was performed in triplicates. **f** Cross reactivity test of cpVNT using antibodies against different viruses or viral antigens spiked in healthy plasma samples. Red line represents a cut-off value at 20%. Each data point represented % inhibition from a single cpVNT experiment. Three separate experiments were performed for each condition.

samples (Supplementary Fig. 9 and Fig. 2c) with an improved LOD of 5 nM, compared to 10 nM in plasma.

Altogether, the cpVNT offers a rapid test, that can be performed in 10 min, for reliable detection of NAbs against SARS-CoV-2. Accuracy of cpVNT, compared to the lab-based methods are well above 80% and 90%, for pVNT and sVNT. The test can be performed both in plasma and serum samples without dilution, therefore facilitating simple workflow at POC settings.

Unlike other reports which predicted that a high level of NAb could be detected from convalescent samples[41], most of the convalescent samples collected for this study showed relatively low NAb level from all three VNT formats tested. The plasma samples collected for this study begun at ~1-month post-admission. Samples containing NAb came mostly from FV1. pVNT, sVNT and cpVNT detected 10, 9 and 9 NAb positive samples, which accounted for 41.7%, 37.5% and 47.4%, of the sample size, respectively. pVNT show 1 and 0 positive samples in FV2 and 3, whereas sVNT and cpVNT, each, showed 2 positive NAbs samples in FV2 and 2 in FV3. Only 3 patients which showed positive NAb status in FV1 completed all 3 visits.

Although a trend of reduction in NAbs is observed in these 3 samples, the sample sizes are too small to determine statistical differences in all test formats. In agreement with our findings, a study from Chia et al., also reported that a substantial number of convalescent samples did not produce sufficient NAb to be detected using sVNT[42]. In addition, a large subgroup of convalescent population showed rapid waning of NAb at 2-month post symptom onset[42], a timeline in which the majority of samples were collected for this study.

## Discussion
Global efforts are underway to improve SARS-CoV-2 surveillance and manage long-term prevention of COVID-19. Assessment of SARS-CoV-2 NAbs is one of the key surveillance criteria required to evaluate herd immunity and the impact of SARS-CoV-2 on a larger population scale. Existing technologies for NAb detection (cVNT, pVNT and sVNT) require laboratory facilities, skilled personnel and long execution times (1 hr–4 days) that are not favorable for large-scale surveillance outside a laboratory setting. cpVNT provides a robust NAb surveillance detection test, that is

simple, rapid and can be easily conducted both inside and outside of laboratories in as little as 10 min. In addition to the cpVNT performance evaluated in this study, it is now feasible to compare its performance to the commercialized sVNT, c-Pass[TM] from GenScript®.

SARS-CoV-2 vaccination programmes have been rolled out in many countries since late 2020. A serological NAb test is a valuable companion test to evaluate the effectiveness of available vaccines. Data from different VNT formats obtained from this study suggested that ~ 37-48% of the samples showed positive NAb status in the first 1-2 months (FV1), post-admission. The number declined to ~7–15% in the subsequent visits. However, due to the small sample size of follow up visits, no statistical difference was observed from different visits. Nonetheless, findings from other studies demonstrated that NAbs declined gradually over 3-month period post symptom onsets[5,43], therefore suggesting that vaccine boosters may be required to maintain the immunity status at a desirable level. The rapid neutralization test described here would be a suitable tool to regularly assess the immune status of individuals, particularly in the vulnerable population. The simple nature and speed of this test provide an accessible POC tool, which can be used at community clinics or in low resource settings, to prioritize vaccine administration. In addition, the test can be rapidly adapted to evaluate the efficiency of NAbs to new virus variants[44–47] and thereby guide the decision-making process in relation to the need for new booster vaccines.

Despite a number of lateral flow assay (LFA) tests available for detection of antibodies against SARS-CoV-2, to our knowledge, only one pre-print report is found for rapid NAbs test[48]. In most LFA antibody detection tests (rapid serology tests), specific antigens are either immobilized on the testing matrix or used as reporting molecules whereas the counter reporting/capturing part is anti-IgA/IgM/IgG antibodies. It was reported by Tan et al.[10] that, when RBD and anti-IgA/IgM/IgG antibodies are used for detection of NAbs in the plate-based ELISA format, non-NAbs are often detected along with NAbs (due to antibodies that bind to RBD but do not possess neutralizing ability)[10]. This method is thus unable to predict the level of NAbs accurately. Applying NAbs test to LFA format seems feasible due to the well-established LFA technology. However, with the test format employed for the cpVNT and sVNT reported in this study, it would be challenging for LFA to report a loss of a colorimetric signal as a positive result, particularly when anti-IgA/IgM/IgG were to be used as reporting molecules. Approximately, 15–20 min incubation time are required for LFA, thus allowing substantial time for non-specific binding to occur at the test line on the LFA test strip. To overcome this issue, a suitable control system would have to be designed to ensure that a positive colorimetric signal generated (lack or low level of NAb) is not due to non-specific binding.

The rapid cpVNT neutralization test developed in this study identifies and measures the very specific interaction between RBD and ACE2 receptor. Non-NAbs will not interfere with the RBD/ACE2 receptor complex formation and the signal detected is specific to NAbs (Figs. 2b, c, 4f). In addition, unlike the LFA format that requires 15–20 min incubation time, the cpVNT requires only 5 min interaction time between NAb/RBD or ACE2/RBD, thus providing minimal time for non-specific interactions and only high-affinity interactions are anticipated to be captured on the cellulose surface.

The sudden high demand for LFA COVID-19 rapid diagnostic tests has created a worldwide shortage of materials required for LFAs, particularly the nitrocellulose membrane, leading to supply chain issues. The cpVNT presented in this study utilizes cellulose membrane which is more economical to produce and supply chains are unimpeded. In addition, cellulose paper can be easily manufactured, enabling large-scale production of the test strips in a very short period, thereby facilitating mass manufacture of the test with low production cost.

Nonetheless, to practically implement the cpVNT as POCT, further developments are required. Plasma/serum is currently optimized for cpVNT. As such, a device capable of separating plasma from whole blood is needed for the POCT applications. Different aspects of test stability are currently being investigated. Based on our preliminary data, with the right preservatives and additives the cellulose test paper can last up to 6 months when kept at ambient temperature (25 °C) without controlling humidity. The test papers last up to 3 years in a desiccator. RBD–CBD and ACE2-BA retain their activities for at least 3 months when kept in optimized conditions. We envision that in the POC settings, a test strip shall comprise one 'Test' spot and one 'Control' spot. The control spot shall host a chemical reaction that indicates the active function of the reagents. The control spot is critical as it ensures that any loss of signal observed is due to the binding of NAb to RBD–CBD and not the malfunction of the chemical reaction. We have explored different optimization parameters for the control spot and found that immobilizing RBD–CBD at high concentrations on the cellulose paper could serve as a control reaction to capture ACE2 tagged to reporting molecules from the assay mixture. Our preliminary data indicated that this approach produced high cyan intensity signals regardless of the presence or absence of NAb in the samples. For signal analysis of cpVNT, we aim to use a pre-set cyan intensity value based on pre-COVID or non-infected samples during the assay optimization stage. This value can be used for the POC applications in which the pre-determined cyan intensity value defines a reference signal for analysis of the cpVNT test results. All these studies are currently being conducted to finalize the test into a usable POCT format.

In conclusion, we have developed a rapid, paper-based cpVNT that can be used at POC for the effective identification of NAbs against COVID19. Comparison of cpVNT against the existing VNTs, including, the pVNT and the sVNT (Fig. 4 and Supplementary Table 3) shows that cpVNT offers a highly competitive solution as compared to existing technologies, including the very rapid execution time (10 min) and ease of operation (no need for laboratory facilities and does not require skilled operators). This represents a significant advance in tackling the pandemic and has far reaching applications.

## Data availability

Source data can be found in Supplementary Data 1. The remaining data can be obtained from the Corresponding authors upon reasonable request.

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

## Acknowledgements

This work is supported by the Singapore Ministry of Health's National Medical Research Council (NMRC) under COVID19RF2-0044 and #R-571-000-081-213 grants, National Health Innovation Centre (NHIC) grant NHIC-COV19-2005004, and National Research Foundation (NRF) via the Campus for Research Excellence and Technological Enterprise (CREATE) funding to the Antimicrobial Resistance Interdisciplinary Research Group (AMR-IRG) of Singapore-MIT Alliance in Research and Technology (SMART) and CREATE Share grant #R571-002-021-592. Single-chain variable fragment (scFv) of the anti-SARS-CoV CR3022 was a kind gift from Prof. Dahai Luo, Lee Kong Chian (LKC) School of Medicine, Nanyang Technology University (NTU). ACE2 stably expressing CHO cells and plasmid encoding SARS-CoV-2 S protein for the pseudotyped lentiviral production, were kind gifts from Assoc Prof. Tan Yee Joo, Department of Microbiology and Immunology, Yong Loo Lin, School of Medicine, National University of Singapore (NUS) and Dr. Conrad Chan, DSO Singapore.

## Author contributions

Conceptualization: P.R.P., H.D.S., P.A.M., M.E.M., P.K., H.J. Conducting VNTs assays: H.L.C., Y.G. B.D/O.S. Protein production and characterization: M.W.C., S.M.L. Supporting in VNTs assay operations: S.Y.N., K.P. Clinical samples collection: P.A.T., H.N., X.G. Processing and storage of clinical samples: R.G., M.M.K., K.P. Assisting in data analysis: D.T., S.K. Funding acquisition: P.R.P., H.D.S., P.K., P.A.M. Writing—original draft: P.K., H.J., Y.G., B.D/O.S. Writing—review and editing: P.R.P., H.D.S., P.K., P.A.M.

## Competing interests

MIT (H.D.S. and S.K.) and SMART (P.K., H.J., and H.D.S.) declare patents and submitted patent applications in relation to this work. P.R.P., P.K., H.J., M.E.M., H.D.S. are co-founder of Thrixen Pte. Ltd a company that aims to explore technology described in this manuscript. All other authors have no competing interests.

**Additional information**

