## [Peer Review File · Communications Medicine]

Reviewers' comments:

Reviewer #1 (Remarks to the Author):

The authors report the development of a cellulose pull-down virus neutralization test (cp-VNT) for rapid point-of-care screening of SARS-CoV-2 neutralizing antibodies. The test were compared with pVNT and sVNT results using COVID-19 convalescent patient plasma/serum. The test takes only 10 min to complete. The work should have impact on the Covid-19 pandemic. However, the experiment was not described clearly, may have fundamental flaws, and needs major revision before publication.

- 1) The main topic of the manuscript is the cpVNT. More details should be provided about the assay, e.g. how many testing regions in a strip, distance between the testing regions, black box geometry for Xiaomi Redmi A9 phone, and what kind of LED. It will be helpful to show a picture of the setup, as well as a photo of the strip taken by the phone.
- 2) in the cpVNT, the authors mixed reagent "A", "B" and plasma together for 5 min. Have the authors tried mixing "A" and plasma first, then with "B" together? What is the rationale to mix everything together? In comparison, in the modified sVNT, neutralizer was incubated with RBD, then ACE2.
- 3) Can author comment on why RGB was converted to CMYK, why only cyan intensity was used and how Cyan intensity was calculated in Sup-Fig 5B?
- 4) How are the "10nM" RBD-CBD and biotin-ACE concentrations determined in the cpVNT assay?
- 5) what is the virus amount (TCID50) in the 25 ul solution in the pVNT assay?
- 6) my understanding of correlation study between pVNT and sVNT should be the correlation of the ID50 values of the neutralization titers of pVNT and sVNT (e.g. in Ref 10). However, fixed dilutions were used: 80x for pVNT and 10x for sVNT. What is the rationale behind it?
- 7) the cut-off for neutralization positive and negative sample should be determined by patient samples with good neutralization antibodies and control. Can the author explain how they reach their value of 50% for pVNT (reference is not clear) and 20% for sVNT?
- 8) Can the authors comment on the sensitivity and specificity of the cpVNT to have real impact on the Covid-19 pandemic?

Reviewer #2 (Remarks to the Author):

This manuscript, entitled "A rapid simple point-of-care assay for the detection of SARS-Cov-2 neutralizing antibodies", is demonstrated a paper-based sensor for SARS-Cov-2 NAb detection to be used POCT. Especially, it was very impressive the well-organized approaches and –characterized convalescent samples to evaluate the developed paper-based sensor. In addition, I was also very

impressed with the relatively high sensitivity and selectivity for NABs on a paper sensor by using ACE receptor and CBD labeled RBD.

I believe this manuscript would be accepted for publication in Communications Medicine after some major revisions as follow;

1. The sensor described in this manuscript has many limitations for use in a resource-limited setting. As a proof-of-concept study, the results should be reliable, but authors should discuss very concretely how to improve the stability and usability.
2. It must discuss the sensor colorimetric signal read-out strategy.
3. How to ensure the long-term stability of the wax printing-based sensors. The wax printing method is a very efficient method to fabricate hydrophobic barriers on paper materials; however, it may not be the final solution. I would like to ask about this issue Authors.
4. The authors mentioned that a lot of optimizations are required to apply the NABs detection to existing LFA, however, in my point of view, it seems to be a design issue of the capture and detection pair, not an issue of the LFA platform itself. And since LFA is a well-established technology in the industry, the current cellulosic paper sensor will have to go through various trials and errors to apply it to the real field.
5. In lines 13-19, page 11, it seems like having some lack of understanding about LFA. It is totally different from research and commercialization. I agree with the author's opinion here, but again in order to use real field, it also required optimization of various factors, for example, use error, ambient humidity/and temperature, lifetime. In terms of these issues, in my opinion, the materials used at the LFA have been verified for at least more than 40 years. However, the sensor used in this study has not been verified others in any way except for performance. So, if there is a discussion about what barriers the proposed sensor needs to overcome to become a real NABs POCT sensor, it will have the more meaningful value of this research one step further.

Reviewer #3 (Remarks to the Author):

The manuscript by Kongsuphol et al describes a method to evaluate the ability of COVID19 convalescent sera to inhibit the binding of labeled-ACE2 to immobilized RBD. Overall, the study is well presented and executed, yet I do not know how novel this concept is. The concept of evaluating the antibody-mediated inhibition of RBD to bind to ACE2 as a correlates to SARS CoV-2 neutralization is not new and was raised and demonstrated before in many manuscripts. Therefore, the only novel findings here are the use of paper-based assay to evaluate the neutralization potency of sera towards SARS CoV-2. In this regard, when compared to other simple binding assays, including ELISA, I'm not sure that this method will provide additional data or will be significantly shorter/easier to use.

Additionally, here are several specific comments:

- Line 147-157 – The authors should provide data and elaborate on the linearity of the binding (w/o antibodies), the LOD etc.
- Line 166 – are these monoclonal Abs? Please provide more data and references for these

antibodies.

- Line 176 – the fact that the LOD is given in concentration of purified antibodies is meaningless when translated to sera. The authors should perform and determine the LOD for diluted sera and express it as the dilution factor.

- Line 187 – please provide the correlation efficiency value.

- Fig2B+C – the data points should be connected with a line and the x-axis should be in log.

- Fig 3 – many points are in negative values which, if I understand correctly, indicate that they actually increased the binding of RBD to ACE2. Is this so? Please add explanation also in the text to these results.

Please find our response to the reviewers comments in 'blue' letters.

Reviewers' comments:

Reviewer #1 (Remarks to the Author):

The authors report the development of a cellulose pull-down virus neutralization test (cp-VNT) for rapid point-of-care screening of SARS-CoV-2 neutralizing antibodies. The test were compared with pVNT and sVNT results using COVID-19 convalescent patient plasma/serum. The test takes only 10 min to complete. The work should have impact on the Covid-19 pandemic. However, the experiment was not described clearly, may have fundamental flaws, and and needs major revision before publication.

1) The main topic of the manuscript is the cpVNT. More details should be provided about the assay, e.g. how many testing regions in a strip, distance between the testing regions, black box geometry for Xiaomi Redmi A9 phone, and what kind of LED. It will be helpful to show a picture of the setup, as well as a photo of the strip taken by the phone.

→ Thank you very much for your constructive instructions. Major revisions, which include detail descriptions of the test construction, chemistry optimization, and the features of the light box, were added to the main text, section "Optimization of rapid paper-based cpVNT", line 144-256.

To further elaborate, detail description of the cellulose test unit construction is explained in the text line 173-189. The text is also quoted here,

"To engineer the vertical flow assay (Fig. 2), wax ink was printed on to cellulose paper to create hydrophobic boundary and define liquid flow path. Liquid flow rate was controlled by stacking 3 layers of cellulose paper together²² (Fig. 2A and D) whereby the top layer has a smaller, 5 mm diameter, of hydrophilic area without wax (testing region) and the lower two layers have larger hydrophilic areas of 6 mm diameters. One unit of cellulose test strip comprises two testing regions for testing of two reactions (Fig. 2A and B). The distance between the center of the two testing spots is 12.5 mm. (Fig. 2A and D). A folded Kim Wipes paper was used as the absorbent pad (Fig. 3A and E). An acrylic sheet with 2 mm thickness was cut using a laser cutter (Epilog Fusion Edge Laser System, USA) into two pieces of an acrylic manifold, each with dimension of 30 x 50 mm² (Fig. 3A, B, C and F). One of the acrylic pieces contains an opening of 25 x 10 mm² (Fig. 2A and C). The other piece of acrylic was equipped with two pieces of 3 mm thickness acrylic sheets which will be used to form spacers for clamping of cellulose test unit (Fig. 2A, B and F). Three layered cellulose papers were stacked on top of the folded Kim Wipes and secured together using the acrylic manifold and paper binders (Fig. 2A and B). The 3 mm spacers between the two pieces of the manifold provide consistent pressure between different cellulose test units prepared. The test strip units provided a consistent liquid flow speed of ~10 sec when 40 μ L of liquid were used.."

Detail description of the light box geometry is explained in the text, line 190-203. The text is also quoted here,

"To capture the colorimetric signal from the cellulose vertical flow assay, Xiaomi Redmi A9 phone was used to captured the image and save in a .jpg format. In order to fix the camera distance and angle as well as to prevent interference from the surrounding light, a 'light box' was produced. The box has a W x L x H dimension of 150 x 230 x 90 mm³ (Fig. 2G). A void was made at the top face of the box to provide an access to the phone camera (Fig. 2H). Distance between cellulose test unit to the camera was 85 mm. Fixtures were made to fix the location for

the test strip unit and the phone whereby the test strip unit holder was secured to the base of the light box (Fig. 2I) and the camera holder was secured to the top face of the light box (Fig. 2G and H). The internal faces of the box were equipped with warm-white LED light strips (Fig. 2I). To prevent shadows on the cellulose test unit, white, cylindrical shaped papers were constructed to cover the LED strip to diffuse light from the LED strips. Example of images captured from the phone camera and the light box are shown in Supplementary Fig. 5D. Images were analyzed using the open source ImageJ software from National Institute of Health (NIH), USA. Only the circular areas of the hydrophilic testing regions were used for analysis.”

The picture of the setup is added to the new Fig. 2. and examples of images captured from the phone camera and light box are shown in the new Supplementary Fig. 5D.

2) in the cpVNT, the authors mixed reagent "A", "B" and plasma together for 5 min. Have the authors tried mixing "A" and plasma first, then with "B" together? What is the rationale to mix everything together? In comparison, in the modified sVNT, neutralizer was incubated with RBD, then ACE2.

→ We appreciated your thoughtful question. We have tried mixing “A” (RBD-CBD) to the plasma sample first, followed by “B” (ACE2-BA/SA-HRP). With this condition, we observed high fluctuations from different replicates. We speculated that this fluctuation may be due to interaction time between NAb and RBD-CBD which is loosely controlled depending on operator speed. The 5 min incubation time only starts when the reagent “B” is applied. To obtain more consistent results, at least 5 min incubation time has to be applied to sample + reagent “A” for the reaction to reach equilibrium. Following the sample + reagent “A” reaction, application of reagent “B” to the reaction would require another 5 min incubation for the second part of the reaction to reach equilibrium. This approach increases total assay time to more than 10 min and introduces more complications to the assay workflow that may not be ideal for POC settings. With an intention to optimize a simplified assay for POC settings, the simpler version of the test which involves only one step incubation of 5 min was selected.

3) Can author comment on why RGB was converted to CMYK, why only cyan intensity was used and how Cyan intensity was calculated in Sup-Fig 5B?

→ Thank you for the question. Please kindly noted that the Sup-Fig 5B is now Sup-Fig 6B. The open source ImageJ software from NIH was used for image analysis. A function of measuring color intensities from different color channels in the RGB space was readily equipped with the software. Building upon the existing tool, a macro which converted RGB color space to CYMK space was manually coded for measurement of color intensities from different channels of the CYMK space. Using different gradients of HRP/TMB blue signal on the cellulose paper, color intensities were measured from different channels from RGB and CYMK spaces. Distinguishable signals were observed with highest ‘difference’ from cyan channel, therefore cyan intensity was selected for image analysis.

This comment is included into the main text line 203-209. The modified text is also quoted here,

“To optimize the image analysis, images were separated into different channels in RGB and CYMK color spaces. Comparing gradient color of the blue signals from HRP/TMB generated on the cellulose paper, signals recorded from cyan channel provided the highest signal (RBD-CBD + ACE2-BA/SA-HRP + TMB) over noise (ACE2-BA/SA-HRP + TMB) ratio as compared to other channels in the RGB and CYMK color spaces. Therefore, the intensity of the cyan channel in the CYMK color space was used for image analysis.”

4) How are the "10nM" RBD-CBD and biotin-ACE concentrations determined in the cpVNT assay?

→ Thank you for your constructive question. Detail optimizations of the RBD-CBD, biotin-ACE2, as well as SA-HRP were added to the main text under “cpVNT optimization” section, line 235-256. New supplementary Fig.7 was also included to clearly demonstrate the results for the optimization process.

The modified text is also quoted here

“Concentrations of RBD-CBD and ACE2-BA/SA-HRP were further optimized to obtain the highest signal difference between presence and absence of NAb (Supplementary Fig. 7). To do so, human serum containing 0 or 100 nM NAb were used for signal comparison. Ratio between maximal and minimal cyan intensities obtained from 0 and 100 nM NAb, respectively, were used to calculate the signal ratio. Concentration of ACE2-BA was first optimized using fixed concentrations of 20 nM and 2 nM of RBD-CBD and SA-HRP, respectively. Signals were captured at 3 min following addition of TMB and washing solution. Results showed that 10 nM ACE2-BA produced highest cyan intensity ratio from 0 nM:100 nM NAb (Supplementary Fig. 7A and B). Therefore 10 nM ACE2-BA was selected for subsequent experiments. To optimize for RBD-CBD concentration, ACE2-BA and SA-HRP concentrations were fixed at 10 nM and 2 nM, respectively. Two different concentrations of RBD-CBD at 10 and 20 nM were tested. Results showed that both concentrations produced similar cyan intensity ratio from 0 nM:100 nM NAb which were at ~2.1. Therefore, different concentrations of NAb were tested to further explore the different cyan intensity signals at various NAb concentrations. This showed that RBD-CBD at 10 nM produced clearer distinguishable cyan intensity signals at lower NAb concentrations (Supplementary Fig. 7C and D) and this concentration was selected for further studies. For optimization of SA-HRP, RBD-CBD and ACE2-BA concentrations were fixed at their optimized concentrations of 10 nM. Various concentrations of SA-HRP were tested in serum containing 0 nM or 100 nM of NAb. Results showed that 6 nM SA-HRP produced highest ratio of cyan intensity from 0 nM: 100 nM NAb (Supplementary Fig. 7E and F). Therefore 6 nM SA-HRP was selected for subsequent analysis. Altogether, optimized concentrations of RBD-CBD, ACE2-BA and SA-HRP for cpVNT were 10 nM, 10 nM and 6 nM, respectively”

5) what is the virus amount (TCID50) in the 25 ul solution in the pVNT assay?

→ We appreciated your thoughtful question. The pseudovirus is diluted to a final concentration of 2×10^6 PFU/ ml. In 25ul there will be 50,000 lentiviral particles. This comment was added to the materials and methods, section “Pseudovirus neutralization assay (pVNT)”, line 528-531.

The modified text and its surrounded context is also quoted here,

“Patient plasma samples were diluted to a final dilution factor of 80 with PBS. The pseudo virus is diluted to a final concentration of 2×10^6 PFU/ ml. In 25ul there will be 50,000 lentiviral particles. The diluted samples were incubated with an equal volume of pseudovirus to achieve a total

volume of 50 μ L, at 37 °C for 1 h.”

6) my understanding of correlation study between pVNT and sVNT should be the correlation of the ID50 values of the neutralization titers of pVNT and sVNT (e.g. in Ref 10). However, fixed dilutions were used: 80x for pVNT and 10x for sVNT. What is the rationale behind it?

→ Thank you for your question. We understand that a comparison of ID50 values would be a more rigorous method to test the correlation between different neutralization tests. In this study, we adopted an alternative approach by testing plasma samples at optimized dilution factors based on past experiences (from previous experiments done during the optimisation phase). When plasma samples from healthy individuals were tested at lower dilution factors (i.e. higher concentration of samples), there was a higher percentage of false positive results from both assays. The 80x and 10x dilution were therefore chosen as the lowest dilution factors that give reliable results from the two assays respectively. The difference in the dilution factors can be attributed to the nature of the assays. The pVNT mimics more physiologically similar conditions in comparison to the sVNT. Thus, a relatively higher dilution factor was required to offset interference.

These comments were elaborated to the main text, line 110-112 for pVNT and 119-120 for sVNT.

The modified texts were also quoted here

“During the test optimization, we found that, a dilution factor of at least 1:80 is necessary to minimize false positives produced by healthy control samples, therefore a fix 1:80 dilution factor is used in this study.”

“10x dilution were chosen as the lowest dilution factors that give reliable results without showing false positives from healthy control samples.”

7) the cut-off for neutralization positive and negative sample should be determined by patient samples with good neutralization antibodies and control. Can the author explain how they reach their value of 50% for pVNT (reference is not clear) and 20% for sVNT?

→ For pVNT, a cut-off of 50% was chosen based on the established pVNT protocols in the reference studies (ref# 13 and 14). According to the reference studies, different plasma dilution factors ranging from 1-10⁶ from healthy control and COVID-19 convalescent samples were used to define a cut-off between presence and absence of NAb. The studies found that a 50% inhibition of the luminescent signal is effective to distinguish between presence and absence of NAb. Therefore, this cut-off is selected for our study. In our assay we found that 1:80 dilution is essential to eliminate false positives generated from healthy control samples. Accordingly, the 1:80 dilution factor is used for our study. To further elaborate, an antibody titer that achieves 50% or higher virus neutralising capacity, is defined as neutralising and below 50% is termed as non-neutralising.

This comment was added to the main text line 103-112

The modified text is also quoted here

“The established pVNT protocols^{13,14} were used as references for pVNT performed in this study. pVNT determined NAb status by measuring chemiluminescent signals from cells infected with pseudovirus. Presence of NAb prevents virus from infecting the cells thereby reducing chemiluminescent signals. Based on the reference studies^{13,14}, a wide range of sample dilution factors from 1–106 were used to determine the effective cut-off between presence and absence of NAb. A 50% signal inhibition was indicated as an effective value to distinguish between presence and absence of NAb. During the test optimization, we found that, a dilution factor of at least 1:80 is necessary to minimize false positives produced by healthy control samples, therefore a fixed 1:80 dilution factor is used in this study.”

For sVNT, a cut-off of 20% was chosen because it is the lowest cut-off that gives an assay specificity of 100% when testing non-neutralizing/neutralizing antibodies spiked in negative plasma (new Supp Fig 3). Using this cut-off, the assay is sensitive to detect RBD-specific neutralizing antibodies at a concentration of 313 ng/mL (2 nM), much lower than the average concentration of these antibodies in COVID-19 convalescent patients.

The text has been modified to further accentuate the description of the cut-off value determination. This text can be found at line 123-128.

The text is also quoted here,

“Receive Operating Characteristic (ROC) curve showed that the test maintained a high sensitivity of 75% while achieving a high specificity of 100% at the inhibitory percentage of ~20%, therefore 20% inhibition is used as a cut-off value to distinguish positive from negative NAb status. Using this cut-off, the sVNT is sensitive to detect RBD-specific neutralizing antibodies at a concentration of 313 ng/mL (2 nM), much lower than the average concentration of these antibodies in COVID-19 convalescent patients¹⁵”

8) Can the authors comment on the sensitivity and specificity of the cpVNT to have real impact on the Covid-19 pandemic?

→ Thank you for your comment. Based on our findings, the cpVNT provides high clinical sensitivity and specificity as compared to pVNT and sVNT. The cpVNT sensitivity and specificity as compared to pVNT are 80% and 84.35%, respectively whereas the sensitivity and specificity as compared to sVNT are 85.71% and 96.55%, respectively. With the rapid 10 min turn-around-time and simple workflow, cpVNT can be operated outside of laboratory settings that is suitable for the population surveillance. The implementation of the current format of the test would provide ease of monitoring of the herd immunity. In particular, following the vaccination programme. The test is feasible as a POC test which could potentially help release social distance regulations. For instance, the test can be used to monitor the protective immune status of travellers, determining the quarantine period, therefore allowing safe opening of the travelling.

Reviewer #2 (Remarks to the Author):

This manuscript, entitled “A rapid simple point-of-care assay for the detection of SARS-Cov-2 neutralizing antibodies”, is demonstrated a paper-based sensor for SARS-Cov-2 NABs detection to be used POCT. Especially, it was very impressive the well-organized approaches and –characterized convalescent samples to evaluate the developed paper-based sensor. In addition, I was also very impressed with the relatively high sensitivity and selectivity for NABs on a paper sensor by using ACE receptor and CBD labeled RBD.

Thank you very much for your comments. We appreciate your positive feedback.

I believe this manuscript would be accepted for publication in Communications Medicine after some major revisions as follow;

1. The sensor described in this manuscript has many limitations for use in a resource-limited setting. As a proof-of-concept study, the results should be reliable, but authors should discuss very concretely how to improve the stability and usability.

→ Thank you very much for you constructive instruction. Discussion on improvement of the test stability and usability have been added to the main text under “Discussion” section, line 359-366

The modified text is also quoted here

“To practically implement the cpVNT as POCT, further developments are required. Plasma/serum is currently optimized for this cpVNT test and a device capable of separating plasma from whole blood is needed for POCT applications. Test stability aspects are currently being investigated and based on our preliminary data with the right preservatives and additives the cellulose test paper can last up to 6 months when kept at ambient temperature (25 ° C) without controlling humidity and 3 years in a desiccator. RBD-CBD and ACE2-BA retain their activities for at least 3 months when kept in optimized conditions. Further studies are currently being conduct to finalize the test into a usable POCT format.”

2. It must discuss the sensor colorimetric signal read-out strategy.

→ Thank you for your comment. Colorimetric readout was done using the open source ImageJ software from NIH. The measurement of the color intensity in RGB space was readily available within the software. Building on the basis of the available feature, a macro was manually added to the software to convert RGB color space to CYMK color space in which the function to measure color intensity from each CYMK channel is the readily extended feature. The readout is carried out only to the circular testing region. Cyan intensity was chose as the measurement tool because it could distinguish the gradient blue color from HRP/TMB with the highest ‘resolution’ as compared to other channels.

Detail on colorimetric signal readout has been added to the main text under “cpVNT optimization” section, line 201-209

The modified text is also quoted here

“Images were analyzed using the open source ImageJ software from National Institute of Health (NIH), USA. Only the circular areas of the hydrophilic testing regions were used for analysis. To optimize the image analysis, images were separated into different channels in RGB and CYMK color spaces. Comparing gradient color of the blue signals from HRP/TMB generated on the cellulose paper, signals recorded from cyan channel provided the highest signal (RBD-CBD + ACE2-BA/SA-HRP + TMB) over noise (ACE2-BA/SA-HRP + TMB) ratio as compared to other channels in the RGB and CYMK color spaces. Therefore, the intensity of the cyan channel in the CYMK color space was used for image analysis.”

3. How to ensure the long-term stability of the wax printing-based sensors. The wax printing method is a very efficient method to fabricate hydrophobic barriers on paper materials; however, it may not be the final solution. I would like to ask about this issue Authors.

→ Thank you very much for your question. In one of our project, we have tested the wax printed paper stability by leaving them in an envelope at room temperature (25 C) without controlling the humidity for more than 6 months, the wax printed papers remain intact through the testing period. In another project, where we have left the wax printed paper on the lab bench with desiccant for ~3 years, the papers remain intact throughout the testing period. With this preliminary data, we are confident that the wax printed paper is stable at ambient condition at least for 6 months and up to 3 years with controlled humidity condition.

This comment has been added to the main text, line 361-364. The text is also quoted here,

“Test stability aspects are currently being investigated and based on our preliminary data with the right preservatives and additives the cellulose test paper can last up to 6 months when kept at ambient temperature (25 °C) without controlling humidity and 3 years in a desiccator.”

4. The authors mentioned that a lot of optimizations are required to apply the NAb detection to existing LFA, however, in my point of view, it seems to be a design issue of the capture and detection pair, not an issue of the LFA platform itself. And since LFA is a well-established technology in the industry, the current cellulosic paper sensor will have to go through various trials and errors to apply it to the real field.

→ Thank you very much for your suggestion. We have modified the text to account for your advice. Applying the NAb test on to the LFA format is practical and feasible, thanks to the well-established LFA technology. However we do anticipate certain challenges. To apply the current test format to LFA, the presence of NAb would lead to reduction in the colorimetric signal on the LFA test strip. We anticipate that this reverse reporting of signal (i.e. negative signals for positive results) may be challenging for test optimization and data interpretation.

This comment has been added to the main text, line 344-349. The modified text is also quoted here

“Adapting a NAb test to a LFA format seems feasible due to the well-established LFA technology. However, with the test format employed for the cpVNT and sVNT reported in this study this would be challenging particular in relation to reporting a loss of a colorimetric signal as a positive result. To overcome this a suitable control system would have to be designed to

ensure that a positive colorimetric signal (i.e. lack or low level of NAb) is not due to non-specific binding.”

5. In lines 13-19, page 11, it seems like having some lack of understanding about LFA. It is totally different from research and commercialization. I agree with the author's opinion here, but again in order to use real field, it also required optimization of various factors, for example, use error, ambient humidity/and temperature, lifetime. In terms of these issues, in my opinion, the materials used at the LFA have been verified for at least more than 40 years. However, the sensor used in this study has not been verified others in any way except for performance. So, if there is a discussion about what barriers the proposed sensor needs to overcome to become a real NAbs POCT sensor, it will have the more meaningful value of this research one step further.

→ Thank you very much for your comment. We have modified the text to address your constructive suggestions. Please refer to response addressed in point. 4 which also addressed your comment in point. 5

We have also added a section to describe necessary studies to realize the cpVNT test to POCT product. Some early stage data have also been described. This description can be found in the main text, line 359-366. The modified text is also quoted here

“To practically implement the cpVNT as POCT, further developments are required. Plasma/serum is currently optimized for this cpVNT test and a device capable of separating plasma from whole blood is needed for POCT applications. Test stability aspects are currently being investigated and based on our preliminary data with the right preservatives and additives the cellulose test paper can last up to 6 months when kept at ambient temperature (25 ° C) without controlling humidity and 3 years in a desiccator. RBD-CBD and ACE2-BA retain their activities for at least 3 months when kept in optimized conditions. Further studies are currently being conduct to finalize the test into a usable POCT format..”

Reviewer #3 (Remarks to the Author):

The manuscript by Kongsuphol et al describes a method to evaluate the ability of COVID19 convalescent sera to inhibit the binding of labeled-ACE2 to immobilized RBD. Overall, the study is well presented and executed, yet I do not know how novel this concept is. The concept of evaluating the antibody-mediated inhibition of RBD to bind to ACE2 as a correlates to SARS CoV-2 neutralization is not new and was raised and demonstrated before in many manuscripts. Therefore, the only novel findings here are the use of paper-based assay to evaluate the neutralization potency of sera towards SARS CoV-2. In this regard, when compared to other simple binding assays, including ELISA, I'm not sure that this method will provide additional data or will be significantly shorter/easier to use.

→ Thank you very much for your valuable comments and suggestions. The key advantages of this technology are (i) the rapid turn-around-time of less than 10 min which is much more rapid than the lab-based method, including ELISA. The cpVNT turn-around-time is also more rapid than other LFA tests which generally take ~15-30 min. (ii) The ease of usage, whereby there will be no liquid waste generated because all reagents are absorbed by the absorbent pad. This advantage make the test suitable for POCT.

Taking into consideration your suggestions, the cpVNT still needs further improvements to realize it as a POCT device. The necessary studies required to make cpVNT a POC test has been modified and added to the main, line 359-366. The modified text is also quoted here

“To practically implement the cpVNT as POCT, further developments are required. Plasma/serum is currently optimized for this cpVNT test and a device capable of separating plasma from whole blood is needed for POCT applications. Test stability aspects are currently being investigated and based on our preliminary data with the right preservatives and additives the cellulose test paper can last up to 6 months when kept at ambient temperature (25 ° C) without controlling humidity and 3 years in a desiccator. RBD-CBD and ACE2-BA retain their activities for at least 3 months when kept in optimized conditions. Further studies are currently being conduct to finalize the test into a usable POCT format.”

Additionally, here are several specific comments:

- Line 147-157 – The authors should provide data and elaborate on the linearity of the binding (w/o antibodies), the LOD etc.

→ Thank you very much for your suggestions. The original text line 147-157 is quoted here

“This simple, vertical flow device takes advantage of the very high affinity of the Cellulose Binding Domain (CBD) for cellulose matrix, causing a rapid immobilization of CBD tagged RBD onto cellulose paper, an event that take place within 10 sec (Supplementary Fig. 5). Moreover, our experiment showed that >80% of RBD/ACE2 receptor complex can be formed within 300 sec (Supplementary Fig. 6B). Based on these observations, the plasma sample is mixed and incubated with the capture reagents for 300 sec to allow efficient formation of RBD-CBD/NAbs or RBD-CBD/ACE2 complexes in aqueous phase before applying it onto the cellulose paper (Supplementary Fig. 5). The high affinity vertical flow assay format ensures that most of the RBD-CBD complexes formed are exposed to the paper matrix and effectively captured through

the rapid and high affinity interaction of CBD and cellulose. The high affinity of CBD to cellulose ensures a minimal loss of the complex during the washing step.”

Kinetic property of RBD-CBD and ACE2 has been elaborated in the BLI study in Supplementary Fig. 5B and C. Using different concentrations of RBD-CBD ranging from 6.25 nM – 100 nM, we found that specific binding between RBD-CBD and ACE2 is observed even as low as 6.24 nM of RBD-CBD. When non-specific target N protein was used, minimal binding between ACE2 and N is observed even at the high 100 nM concentration. Low K_D value of 12.7 nM was observed between RBD-CBD and ACE2 from our BLI study. This result confirmed that the two proteins have high binding affinity toward each other.

Majority part of the manuscript has been modified to explain detail description of the cpVNT construction and optimization. Text has also been modified to highlight the kinetic property of RBD-CBD and ACE2. This information can be found in the main text line 162-172

The modified text is also quoted here,

“To construct the test, recombinant (i) RBD-CBD and (ii) biotin (BA) tagged monoFc-ACE2 receptor proteins were expressed, purified (Supplementary Fig. 5A) and evaluated for their kinetic properties using bio-layer interferometry (BLI). Various concentrations of RBD-CBD ranging from 6.25 nM – 100 nM were used. Data showed specific binding between monoFc-ACE2 and RBD-CBD event at a low concentration of RBD-CBD at 6.25 nM (Supplementary Fig. 5B). The pair demonstrated high affinity toward each other with a K_D of 12.7 nM (Supplementary Fig. 6B) which is comparable to previously reported K_D ranging from 4.7–15.2 nM¹⁸⁻²¹. In contrast, SARS-CoV-2 structural, nucleocapsid (N) protein was tested on ACE2-BA to assess specificity of the protein. BLI data showed minimal interaction of BA-ACE2 with N protein even at 100 nM of N, indicating that recombinant BA-ACE2 is specific for RBD protein (Supplementary Fig. 5C).”

- Line 166 – are these monoclonal Abs? Please provide more data and references for these antibodies.

→ Thank you for your comment. Yes, mouse anti SARS-CoV-2 NAbs (cat# 40591-MM43-100) from Sino Biological, USA was used for the test optimization. The information has been provided in the Material and method section line 405-409.

- Line 176 – the fact that the LOD is given in concentration of purified antibodies is meaningless when translated to sera. The authors should perform and determine the LOD for diluted sera and express it as the dilution factor.

→ Thank you very much for your comment. We aim to develop the cpVNT for POCT application. Therefore we opted for using different concentrations of antibody instead of plasma dilution factor because it represents the POCT format we have envisioned. To elaborate further, for cpVNT, plasma or other patient sample won't be further processed prior to applying it onto the test. With this objective in mind, plasma containing different concentrations of NAb were used. The representation of LOD in antibody concentration format (nM) depicted the initial part of the test optimization. To accommodate for test form factor, the final test report is designed to be presented as % inhibition. Drawing of the cut-off for % inhibition has been subsequently demonstrated in Fig 4A.

Your comment holds a valid point and is valuable to us. To address your comment, we used the lab-based pVNT and sVNT as reference tests in comparison to our test. Both of these lab-based tests have been thoroughly optimized and established and have reference standards against dilution factors. We validated our test using these two lab-based methods and employed the test sensitivity, specificity and accuracy to determine our test performance.

- Line 187 – please provide the correlation efficiency value.

→ Thank you for your constructive comment. Pearson correlation coefficients have been added to the main text at line 281-282

The modified text is also quoted here,

“Data obtained from cpVNT show a high correlation with pVNT and sVNT with Pearson correlation coefficients of 0.7 and 0.87 respectively.”

- Fig2B+C – the data points should be connected with a line and the x-axis should be in log.

→ Thank you for your constructive suggestions. The previous Fig. 2B and C are now Fig. 3B and C. Figures have been modified accordingly.

- Fig 3 – many points are in negative values which, if I understand correctly, indicate that they actually increased the binding of RBD to ACE2. Is this so? Please add explanation also in the text to these results.

→ We appreciate your comment. Text has been added accordingly, line 273-275

The added text is also quoted here,

“Negative inhibitory percentages were observed from some data points. The negative values were due to the higher cyan signals as compared to the pre-COVID negative control samples. In the context of cpVNT, it can be interpreted that no NAbs were detected from these samples”

** See Nature Research's author and referees' website at <https://ddec1-0-en-ctp.trendmicro.com:443/wis/clicktime/v1/query?url=www.nature.com%2fauthors&umi=d=cad968d3-f4c5-491e-ab43-ea89054cd772&auth=d0d2ee656b1a8c5d3181c6674d023d4ed062f583-b564d1fda19a9bbb9da44b12ad0b5d3d40508a06> for information about policies, services and author benefits

Reviewers' comments:

Reviewer #1 (Remarks to the Author):

The authors have made improvement on the clarity of the manuscript. However, there are still places that need attention before its publication.

To summarize the manuscript "Results" points, the authors are reporting a novel paper-based SARS-CoV-2 cpVNT POC test based on vertical flow format and strong affinity of CBD to paper. After optimization, standard curve was constructed using spiked known NAb, while non-Nabs showed background neutralization, with a LOD of 10 nM NAb. Using COVID-19 convalescent plasma and control plasma, a cutoff for existence of NAb was determined to be 20%, which showed most NAb positive samples in FV1.

To compare cpVNT with more conventional methods, modified pVNT and sVNT were developed based literature. For pVNT, a fixed 80x sample dilution was used. The cutoff for existence of NAb was set by 50%, same as in the literature. For sVNT, the assay was modified with RBD immobilized on the plate instead of ACE2. A fixed 10x sample dilution was used to avoid false positives from healthy controls. Known NAb and non-NAb were spiked in plasma to generate the dose response curves. 20% was selected based on ROC with 75% sensitivity and 100% specificity. After pVNT and sVNT, correlations among cpVNT, sVNT and pVNT were generated to be cpVNT-pVNT 0.70, cpVNT-sVNT 0.87, and sVNT-pVNT 0.8. Using pVNT as a "gold" standard, the cpVNT showed sensitivity and specificity of 80% and 84.4%, and accuracy of 83.3%.

The authors also characterized the immunological profile of the COVID-19 convalescent plasma for IgA, IgM and IgG levels.

The following questions need to be addressed:

- 1) In the abstract, the authors pointed out that the cpVNT relies on two key technologies: a) vertical flow paper-based assay; b) rapid interaction of CBD to cellulose paper. However, there is not much introduction on the two technologies. The authors should cite references on the two technologies to give readers more background info on the two technologies. For example, for vertical flow paper-based assay, the author could cite "Hsu et al., *Biomaterials* 35 (2014) 3729-3735", which showed quantitative paper-based ELISA, and "Chen et al., *Talanta* 191 (2019) 81-88", which showed the advantage of vertical flow assay with a thin paper thickness as the flow path that allowed fast sample flow through the paper even with nanoscale pore size. Similarly, more background/references should be given for CBD.
- 2) The "Results" of the manuscript started with the characterization of the COVID-19 convalescent plasma for IgA etc., which is not the focus of the manuscript. It is suggested the manuscript be structured as the "Summarized results" above to make the manuscript flow better.
- 3) The cpVNT assay was optimized by maximizing the signal ratio between 0 and 100 nM NAb. Suggest to remove Fig. 3B because it is essentially the same as Fig 3C. It is also suggested that a full sigmoidal dose-response curve can be shown and fitted with the 4-parameter logistic model. A good R^2 value can help to validate the assay, as well as determine the high- low- signal for the assay to reduce negative inhibition percentage values. Plus, what is the strategy of selecting the "cut-off" values to be 20%? To keep the control to be NAb negative, the "cut-off" can be as low as 12%?
- 4) pVNT was used as the "gold" standard for determining existence of NAb. However, this "pVNT" is different from the traditional "pVNT" cited (Ref# 14) in that it didn't measure titer to reach 50%

neutralization, but use a fixed dilution to measure neutralization value. Does this mean that this “pVNT” is mainly for qualitative characterization of NABs (i.e. just Yes or No) to establish a standard for the cpVNT, instead of quantitative assay of the NABs? Plus, only 50K PFU was used per well, which is only 1.4x of TCID50 (TCID50 estimated by $1.4 \times 50K/2$, where 50K cells seeded per well). Group of Ref#14 used 650 TCID50/well, and showed low virus# (< 163 TCID50/well) didn't give good dose response curve (Nie et al., Emerging Microbes & Infections (2020) Vol. 9 <https://doi.org/10.1080/22221751.2020.1743767>). Can the author explain why such a low virus# was used, and show a dose response curve using the known NAB for the “pVNT” assay used here, with 4-parameter logistic curve fitting?

5) For the modified sVNT assay, a dose response curve with 4-parameter logistic fitting and R^2 value is also recommended.

6) Can the authors explain the format of cpVNT POC assay in real application? Will there be still two spots, one used for the sample, another used for a reagent control? Then the neutralization percentage is calculated by the sample Cyan intensity and some pre-established Cyan intensity of negative control? Or the Cyan intensity of negative control will be generated together with the sample Cyan intensity? How many control spots will the assay have?

7) VNTs of cpVNT and possible LFA should both reporting a loss of colorimetric signal as a positive result due to the competitive format. Not sure why the authors say it would be challenging only for LFA.

8) The COVID-19 convalescent samples showed only a small portion to be NAB positive, which is different from recent report that NAB levels are highly predictive of immune protection from SARS-CoV-2 infection with Convalescent sample protection efficacy above 50% for months (Khoury et al., Nature Medicine (2021) <https://doi.org/10.1038/s41591-021-01377-8>). can this be due to the selection of your assay condition for cutoff?

9) Correct typos within the manuscripts.

Reviewer #2 (Remarks to the Author):

I think the revised manuscript reflects the answer to my question well. I accept this revised manuscript.

Reviewer #3 (Remarks to the Author):

The authors have addressed all my comments

We are very happy that reviewers 2 and 3 find the manuscript suitable for publication and are satisfied with the responses provided.

We have now addressed the remaining issues pointed out by reviewer 1.
Reviewers' comments:

Reviewer #1 (Remarks to the Author):

The authors have made improvement on the clarity of the manuscript. However, there are still places that need attention before its publication.

To summarize the manuscript "Results" points, the authors are reporting a novel paper-based SARS-CoV-2 cpVNT POC test based on vertical flow format and strong affinity of CBD to paper. After optimization, standard curve was constructed using spiked known NAb, while non-Nabs showed background neutralization, with a LOD of 10 nM NAb. Using COVID-19 convalescent plasma and control plasma, a cutoff for existence of NAb was determined to be 20%, which showed most NAb positive samples in FV1.

To compare cpVNT with more conventional methods, modified pVNT and sVNT were developed based literature. For pVNT, a fixed 80x sample dilution was used. The cutoff for existence of NAb was set by 50%, same as in the literature. For sVNT, the assay was modified with RBD immobilized on the plate instead of ACE2. A fixed 10x sample dilution was used to avoid false positives from healthy controls. Known NAb and non-NAb were spiked in plasma to generate the dose response curves. 20% was selected based on ROC with 75% sensitivity and 100% specificity. After pVNT and sVNT, correlations among cpVNT, sVNT and pVNT were generated to be cpVNT-pVNT 0.70, cpVNT-sVNT 0.87, and sVNT-pVNT 0.8. Using pVNT as a "gold" standard, the cpVNT showed sensitivity and specificity of 80% and 84.4%, and accuracy of 83.3%.

The authors also characterized the immunological profile of the COVID-19 convalescent plasma for IgA, IgM and IgG levels.

The following questions need to be addressed:

1) In the abstract, the authors pointed out that the cpVNT relies on two key technologies: a) vertical flow paper-based assay; b) rapid interaction of CBD to cellulose paper. However, there is not much introduction on the two technologies. The authors should cite references on the two technologies to give readers more background info on the two technologies. For example, for vertical flow paper-based assay, the author could cite "Hsu et al., Biomaterials 35 (2014) 3729-3735", which showed quantitative paper-based ELISA, and "Chen et al., Talanta 191 (2019) 81-88", which showed the advantage of vertical flow assay with a thin paper thickness as the flow path that allowed fast sample flow through the paper even with nanoscale pore size. Similarly, more background/references should be given for CBD.

Author response: Thank you very much for your instructive comments and guidance. We have included related references to the vertical flow assay, including

ref.#11-14 and usage of CBD that enables assay reaction to take place on the cellulose vertical flow assay, including ref #14-17.

Modified text is shown in blue face text line **#84-92**. The modified text is also quoted here.

Vertical flow assay formats allow reagents to flow in a top-to-bottom fashion for detection of biomolecules. Vertical flow assays offer short liquid flow paths that enable rapid and controllable flow speed for handling of liquid reagent¹¹⁻¹⁴. With an aim for a POC NAb test, the vertical flow assay format with cellulose as a cost-effective material that can easily be manufactured at scale was selected for this study. Here we demonstrate that cellulose can be used as a test matrix for vertical flow assays, where the assay reaction is occurring on the cellulose matrix using high affinity interactions between the cellulose binding domain (CBD) and the cellulose matrix^{14,15}. The common construction of the assay is to fuse CBD to the capture reagent¹⁴⁻¹⁷.

2) The “Results” of the manuscript started with the characterization of the COVID-19 convalescent plasma for IgA etc., which is not the focus of the manuscript. It is suggested the manuscript be structured as the “Summarized results” above to make the manuscript flow better.

Author response: Thank you for your instruction. We have re-arranged the story to highlight the construction of cpVNT prior to reporting characterization of COVID-19 convalescent plasma. The results are now structured as follow:

- (i) Optimization of rapid paper-based cpVNT
- (ii) The rapid paper-based cpVNT performance
- (iii) Assessment of SARS-CoV-2 immunological profile from COVID-19 convalescent plasma
- (iv) Evaluating of cpVNT performance using COVID-19 convalescent plasma.

3) The cpVNT assay was optimized by maximizing the signal ratio between 0 and 100 nM NAb. Suggest to remove Fig. 3B because it is essentially the same as Fig 3C. It is also suggested that a full sigmoidal dose-response curve can be shown and fitted with the 4-parameter logistic model. A good R² value can help to validate the assay, as well as determine the high- low- signal for the assay to reduce negative inhibition percentage values. Plus, what is the strategy of selecting the “cut-off” values to be 20%? To keep the control to be NAb negative, the “cut-off” can be as low as 12%?

Author response: We appreciate your comments. Original Fig. 3B plot (cyan intensity signals) was removed. As the storyline was re-constructed the original Fig. 3 is now Fig. 2. The new Fig 2C is replaced with a full sigmoidal dose-response curve fitted with 4-parameter logistic model as per recommendation. R² was determined to be 0.9534. Please see the modified Fig. 2 at line **#786**.

The 20% cut-off for cpVNT was selected to ensure that the test produces no false positive results. Mean \pm SD of the control group was -0.63 ± 18.6 % inhibition. Therefore to accommodate for the standard deviation, we selected 20% inhibition as the cut-off value.

Results which showed negative inhibition were due to higher cyan intensity as compared to the mean value of the control group (pre-COVID + non-infected

samples). In this context, negative percentage is considered as no detection of nAb. This comment is included in the main text line #285-288 to further clarify the test results. The modified text is also quoted here.

Negative inhibitory percentages were observed from some data points. The negative values were due to the higher cyan intensity signals as compared to the reference point calculated from mean value of pre-COVID and non-infected samples. In the context of cpVNT, it can be interpreted that no NABs were detected from these samples.

4) pVNT was used as the “gold” standard for determining existence of NABs. However, this “pVNT” is different from the traditional “pVNT” cited (Ref# 14) in that it didn’t measure titer to reach 50% neutralization, but use a fixed dilution to measure neutralization value. Does this mean that this “pVNT” is mainly for qualitative characterization of NABs (i.e. just Yes or No) to establish a standard for the cpVNT, instead of quantitative assay of the NABs?

Plus, only 50K PFU was used per well, which is only 1.4x of TCID50 (TCID50 estimated by $1.4 \times 50K/2$, where 50K cells seeded per well). Group of Ref#14 used 650 TCID50/well, and showed low virus# (< 163 TCID50/well) didn’t give good dose response curve (Nie et al., Emerging Microbes & Infections (2020) Vol.9 <https://doi.org/10.1080/22221751.2020.1743767>). Can the author explain why such a low virus# was used, and show a dose response curve using the known NAB for the “pVNT” assay used here, with 4-parameter logistic curve fitting?

Author response: Thank you for your comments. Several versions of pVNT assays have been utilized to provide a qualitative read-out plus degree of quantitation of neutralizing antibody titres as a substitute to the traditional Plaque Reduction Neutralization Test (PRNT) and in-vitro virus neutralisation assays, using WT SARS-CoV-2 virus. In addition to the original cited reference (originally ref. #14 (Nie et al.), now ref. #27), more references describing different versions of pVNT assays are cited in this revised manuscript including ref. #26-31.

Whilst pVNT assay can be viewed as largely qualitative in nature, it does provide a quantitative comparator between samples containing neutralizing antibodies. This is evidence by the high correlation between live virus neutralisation assays and the pVNT assays (ref #24-25). Moreover, a read-out of neutralization percentages from live virus neutralisation assays has been correlated with protection from COVID disease (ref# 35). Altogether, these literatures suggested that pVNT represents concordance results to live virus and can be, in general, used, as quantitative assay to determine nAb activity.

The method for pVNT assay utilized in this study (including generation of SARS-CoV-2 pseudotyped lentivirus, type of cell lines, cell density and pseudovirus concentration) was optimized based on the published study: *Two linear epitopes on the SARS-CoV-2 spike protein that elicit neutralising antibodies in COVID-19 patients by Poh et.al. (ref#30)*. This information has been clarified in the revised manuscript.

Based on the established protocol reported in Poh et al. (ref#30), % Neutralization of NAb titer obtained from COVID-19 convalescent plasma were plotted and fitted to

the 4 parameter logistic model curves. During the pVNT optimization for this study, we re-validated the assay using COVID-19 convalescent plasma. This data is shown in **Fig. R1**. We also tested the optimized pVNT used in this study with recombinant neutralizing human antibodies that are currently undergoing clinical development (Wang et.al¹, Chi et.al² and Brouwer et.al³). Titration curves from the recombinant neutralizing antibodies were also plotted and fitted to the 4 parameter model curves as shown in **Fig. R2**. These optimized data demonstrated that the pVNT performed well in convalescent samples and recombinant neutralizing antibodies. Based on this optimized data and the published data from Poh et al., we simplified the pVNT by fixing the sample dilution to 1:80. At this dilution, the assay can clearly distinguish NAb activity from different convalescent samples at 50% neutralization. pVNT optimization is not the main focus of this study. As such, these additional data are not included in the manuscript because they were part of our assay optimization and have been published by Poh et al (ref# 30).

Although pVNT can be used to quantitatively identify NAb activity, we used the 50% neutralization of pVNT as a qualitative cut-off to define the clinical sensitivity, specificity, and accuracy of the developed cpVNT. To determine these parameters, qualitative test interpretation is required to set reference points for 'true positives', 'false positives', 'true negatives', and 'false negatives', whereby the results from pVNT were set as reference for 'true positives' and 'true negatives'.

Neutralisation titre curves for in-house convalescent plasma samples

Fig. R1. Pseudovirus neutralisation assay conducted on SARS-CoV2 convalescent plasma samples titrated at a range of dilution factors.

¹Wang B et al. Bivalent binding of a fully human IgG to the SARS-CoV-2 spike proteins reveals mechanisms of potent neutralization. *bioRxiv* (2020). <https://doi.org/10.1101/2020.07.14.203414>

² Chi X et al. A neutralizing human antibody binds to the N-terminal domain of the Spike protein of SARS-CoV-2. *Science*. 369, 650-655 (2020). 10.1126/science.abc6952

³ Brouwer PJM et al. Potent neutralizing antibodies from COVID-19 patients define multiple targets of vulnerability. 369, 643-650 (2020). 10.1126/science.abc5902

Neutralisation titre curves for published Antibodies

Fig R2. Pseudovirus neutralisation assay conducted on SARS-COV2 recombinant human neutralising antibodies titrated at a range of concentrations

Text has been modified accordingly at line #234-248 to clarify the pVNT optimization. The modified text is also quoted here

pVNT provides results that are closely correlated to conventional virus neutralization test using live virus^{24,25}. Therefore it is widely used as a substitute method for conventional Plaque Reduction Neutralization Test (PRNT) and in-vitro live virus neutralization assays, to quantitatively assess NAb status²⁶⁻³¹. The established pVNT protocols³⁰ was used as a reference for pVNT performed in this study. pVNT determined NAb status by measuring chemiluminescent signals from cells infected with pseudovirus. Presence of NABs prevent virus from infecting the cells thereby reducing chemiluminescent signals. Based on the test optimization and the reference study³⁰, a wide range of sample dilution factors from 10¹-10⁶ were used to determine the effective cut-off between presence and absence of NAb. A 50% signal inhibition was indicated as an effective value to distinguish between presence and absence of NAb. Similarly, pVNT established by different research groups also demonstrate 50% signal inhibition as an effective cut-off. During the test optimization, we found that, a dilution factor of at least 1:80 is necessary to minimize false positives produced by healthy control samples, therefore a fix 1:80 dilution factor is used in this study as a simplified pVNT.

5) For the modified sVNT assay, a dose response curve with 4-parameter logistic fitting and R² value is also recommended.

Author response: We appreciated your constructive comment. sVNT dose response was modified to sigmoidal curve with 4 parameter logistic fitting. R² value was calculated to be 0.9901. Please see the modified **Supplementary Fig. 6A** at the supplementary information line #130.

6) Can the authors explain the format of cpVNT POC assay in real application? Will there be still two spots, one used for the sample, another used for a reagent control? Then the neutralization percentage is calculated by the sample Cyan intensity and

some pre-established Cyan intensity of negative control? Or the Cyan intensity of negative control will be generated together with the sample Cyan intensity? How many control spots will the assay have?

Author response: Thank you for your instructions. Our vision of creating the POC assay aligns well with you advice. The test shall comprises a 'Test' and a 'Control' spots, where the control spot serves to ensure active chemistry function and validate the loss of signals observed from the test spot upon presence of NAb in the sample. We further discussed a potential design of control spot whereby we found that immobilization of RBD-CBD at high concentrations on the cellulose surface can be used as a control spot reaction to capture ACE2 tagged to reporting molecules from the assay mixture. We found that with this design, high cyan intensity signals can be observed from the assay mixture regardless of presence or absence of NAb.

Pre-established cyan intensity obtained from pre-COVID and non-infected samples can be used as a pre-defined value to analyze signals from cpVNT. We are currently working to define this value for the POC application.

These responses are included in the main text, line #393-405. The modified text is also quoted here.

We envision that in the POC settings, a test strip shall comprise one 'Test' spot and one 'Control' spot, whereby the control spot shall host a chemical reaction that indicates active function of the reagents. The control spot is critical as it ensures that any loss of signal observed is due to binding of NAb to RBD-CBD and not the malfunction of the chemistry reaction. We have explored different optimization parameters for the control spot and found that immobilizing RBD-CBD at high concentrations on the cellulose paper could serve as a control reaction to capture ACE2 tagged to reporting molecules from the assay mixture. Our preliminary data indicated that this approach produced high cyan intensity signals regardless of presence or absence of NAb in the samples. For signal analysis of cpVNT, we aim to use a pre-set cyan intensity value based on pre-COVID or non-infected samples during the assay optimization stage. This value can be used for the POC applications in which the pre-determined cyan intensity value defines a reference signal for analysis of the cpVNT test results. All these studies are currently being conducted to finalize the test into a usable POCT format.

7) VNTs of cpVNT and possible LFA should both reporting a loss of colorimetric signal as a positive result due to the competitive format. Not sure why the authors say it would be challenging only for LFA.

Author response: Thank you for your question. Most established LFA utilizes anti IgA/IgM/IgG as the capture or reporter molecules, which often leads to non-specific binding. Using this format of LFA, we anticipated that non-specific bindings from anti-IgA/IgM/IgG would interfere with the positive signals, in which the non-specific binding may lead to unwanted signal observed from the test line. This response is further clarify in the main text line #356-379. The modified text is also quoted here.

Despite a number of lateral flow assay (LFA) tests available for detection of antibodies against SARS-CoV-2, to our knowledge, only one pre-print report is found for rapid NAb test⁴². In most LFA antibody detection tests (rapid serology tests), specific antigens are either immobilized on the testing matrix or used as reporting molecules whereas the counter reporting/capturing part are anti-IgA/IgM/IgG antibodies. It was reported by Tan et al¹⁰ that, when RBD and anti-

IgA/IgM/IgG antibodies are used for detection of NAb in the plate-based ELISA format, non-NAb are often detected along with NAb (due to antibodies that bind to RBD but do not possess neutralizing ability)¹⁰. This method is thus unable to predict the level of NAb accurately. Adapting a NAb test to a LFA format seems feasible due to the well-established LFA technology. However, with the test format employed for the cpVNT and sVNT reported in this study, it would be challenging for LFA to report a loss of a colorimetric signal as a positive result, particularly when anti-IgA/IgM/IgG were to be used as reporting molecules. Approximately, 15-20 minutes incubation time are commonly required for LFA, thus allowing substantial time for non-specific binding to occur at the test line on the LFA test strip. To overcome this issue, a suitable control system would have to be designed to ensure that a positive colorimetric signal generated (i.e. lack or low level of NAb) is not due to non-specific binding.

The rapid cpVNT neutralization test developed in this study identifies and measures the very specific interaction between RBD and ACE2 receptor. Non-NAb will not interfere with the RBD/ACE2 receptor complex formation and the signal detected is specific to neutralizing antibodies (Fig. 2B-C and 4F). In addition, unlike the LFA format that requires 15-20 min incubation time, the cpVNT developed in this study requires only 5 min interaction time between NAb/RBD or ACE2/RBD, thus providing minimal time for non-specific interactions and only high affinity interactions are anticipated to be captured on the cellulose surface.

8) The COVID-19 convalescent samples showed only a small portion to be NAb positive, which is different from recent report that NAb levels are highly predictive of immune protection from SARS-CoV-2 infection with Convalescent sample protection efficacy above 50% for months (Khoury et al., Nature Medicine (2021) <https://doi.org/10.1038/s41591-021-01377-8>). can this be due to the selection of your assay condition for cutoff?

Author response: Thank you for your question. The low NAb signals were observed from all three types of VNTs tested in this study. All three VNTs were performed independently in a blind testing fashion. Data obtained from the three independent assays confirmed similar trend of results.

We speculated that the low NAb results may be due to the clinical samples recruited for this study. A study from Chia et al., (Lancet (2021), 2, e244, DOI:[https://doi.org/10.1016/S2666-5247\(21\)00025-2](https://doi.org/10.1016/S2666-5247(21)00025-2)) reported that substantial number of COVID-19 convalescent samples did not produce sufficient level for NAb to be detected using sVNT. In addition, a large subgroup of convalescent samples showed rapid waning of NAb follow 2 months of symptom onset, the timeline which corresponded to sample collected for our study. With this supporting information and test results obtained from three different VNTs performed in our lab, we speculated that the convalescent samples recruited for our study generally present low NAb status.

This comments were included in the main text line **#315-328**. The modified text is also quoted here.

Unlike other reports which predicted that high level of NAb could be detected from convalescent samples³⁵, most of the convalescent samples collected for this study showed relatively low NAb level from all three VNT formats tested. The plasma samples collected for this study begun at ~1 month post admission. Samples containing NAb came mostly from FV1. pVNT, sVNT and cpVNT detected 10, 9 and 9 NAb positive samples, which accounted for 41.7%, 37.5% and 47.4%, of the sample size, respectively. pVNT show 1 and 0 positive samples in FV2 and 3, whereas sVNT and cpVNT, each, showed 2 positive NAb samples in FV2 and 2 in FV3. Only 3

patients which showed positive NAb status in FV1 completed all 3 visits. Although a trend of reduction in NAbs is observed in these 3 samples, the sample sizes are too small to determine statistical differences in all test formats. In agreement with our findings, a study from Chia et al., also reported that substantial number of convalescent samples did not produce sufficient NAb to be detected using sVNT³⁶. In addition, a large subgroup of convalescent population showed rapid waning of NAb at 2-month post symptom onset³⁶, a timeline in which the majority of samples were collected for this study.

9) Correct typos within the manuscripts.

Author response: Thank you for your comments. We have reviewed the manuscript thoroughly and corrected all the identified typos.

Reviewer #2 (Remarks to the Author):

I think the revised manuscript reflects the answer to my question well. I accept this revised manuscript.

Reviewer #3 (Remarks to the Author):

The authors have addressed all my comments

REVIEWERS' COMMENTS:

Reviewer #1 (Remarks to the Author):

The authors have addressed my comments. Please correct one mistake:

1) Line 102, TMB/H₂O₂ doesn't hydrolyze HRP to generate the color. please correct the mechanism of the HRP-TMB-H₂O₂ colorimetric detection.

No further review is needed from me.

We have addressed the minor comment by reviewer 1:

- 1) *Line 102, TMB/H₂O₂ doesn't hydrolyze HRP to generate the color. please correct the mechanism of the HRP-TMB-H₂O₂ colorimetric detection.*

To now state: "A complex of ACE2-BA/SA-HRP was used to generate colorimetric signal via application of 3,3',5,5'-tetramethylbenzidine (TMB)/H₂O₂, in which HRP catalyzes oxidation of TMB substrate, producing blue color signals."